# Miss-ReID: Delivering Robust Multi-Modality Object Re-Identification Despite Missing Modalities

**Ruida Xi**
State Key Laboratory of Electromechanical Integrated
Manufacturing of High-Performance Electronic Equipment, Xidian University
`ruidaxi@stu.xidian.edu.cn`

## Abstract

Multi-modality object Re-IDentification (ReID) targets to retrieve special objects by integrating complementary information from diverse visual sources. However, existing models that are trained on modality-complete datasets typically exhibit significantly degraded discrimination during inference with modality-incomplete inputs. This disparity highlights the necessity of developing a robust multi-modality ReID model that remains effective in real-world applications. For that, this paper delivers a flexible framework tailored for more realistic multi-modality retrieval scenario, dubbed as **Miss-ReID**, which is the first work to friendly support both the modality-missing training and inference conditions. The core of Miss-ReID lies in compensating for missing visual cues via vision-text knowledge transfer driven by Vision-Language foundation Models (VLMs), effectively mitigating performance degradation. In brief, we capture diverse visual features from accessible modalities first, and then build memory banks to store heterogeneous prototypes for each identity, preserving multi-modality characteristics. Afterwards, we employ structure-aware query interactions to dynamically distill modality-invariant object structures from existing localized visual patches, which are further reversed into pseudo-word tokens that encapsulate the identity-relevant structural semantics. In tandem, the inverted tokens, integrated with learnable modality prompts, are embedded into crafted textual template to form the personalized linguistic descriptions tailored for diverse modalities. Ultimately, harnessing VLMs' inherent vision-text alignment capability, the resulting textual features effectively function as compensatory semantic representations for missing visual modalities, after being optimized with some memory-based alignment constraints. Extensive experiments demonstrate our model's efficacy and superiority over state-of-the-art methods in various modality-missing scenarios, and our endeavors further propel multi-modality ReID into real-world applications.

## 1  Introduction

Object Re-Identification (ReID) aims to retrieve specific objects, such as pedestrians, vehicles and other trackable entities, across non-overlapping cameras. Despite remarkable progress in traditional RGB-based single-modality object ReID over recent decades [1–7], its robustness remains compromised in complex environments, such as low light and varying weather. Fortunately, multi-modal imaging technologies emerge as a promising solution, effectively mitigating the limitations of single-modality ReID by integrating complementary information from diverse visual sources such as RGB, Near Infrared (NIR) and Thermal Infrared (TIR) modalities. Consequently, multi-modality object ReID methods have garnered significant attention in this filed [8–15].

39th Conference on Neural Information Processing Systems (NeurIPS 2025).

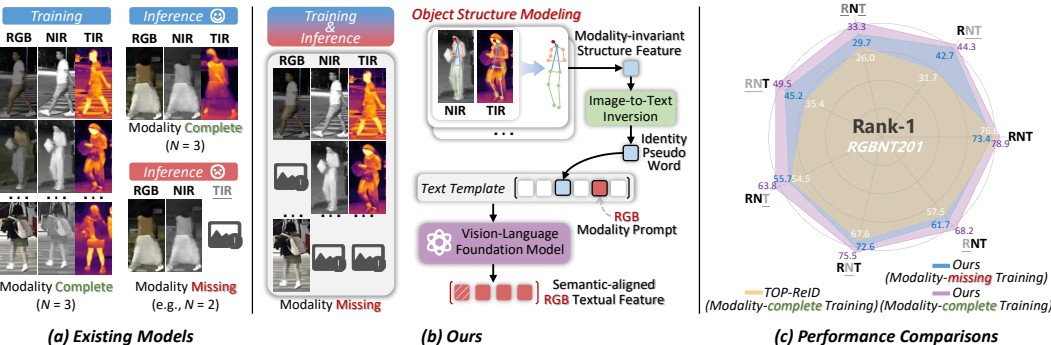

*(a) Existing Models*     *(b) Ours*     *(c) Performance Comparisons*

Figure 1: Illustrative comparisons between existing multi-modality object ReID methods and ours. (**a**) Prior works, when trained on modality-complete datasets, typically exhibit robust performance during inference under modality-complete conditions, while showing degraded performances when encountered with modality-missing cases in practice. Here, $N$ denotes the number of available modalities. (**b**) Our work studies a more general scenario, where various modality-missing cases would occur at both training and inference. (**c**) Performance comparisons between TOP-ReID [9] and Miss-ReID (Ours) on the RGBNT201 benchmark. Abbreviations: **R**: **R**GB; **N**: Near Infrared; **T**: Thermal Infrared. **RNT** indicates that evaluating the modality-complete data during **inference**, **RNT** excludes RGB images from the evaluation, and others omit specific modalities in analogous patterns.

Though exhibiting promising performance, existing multi-modality ReID methods [16, 11, 13, 12, 17] typically rely on an assumption regarding the modality completeness of data, which may not hold in practice owing to privacy protections, sensor failures or security requirements [18]. Specifically, as illustrated in Fig. 1 (a), previous multi-modal ReID models, when trained on modality-complete datasets, exhibit robust performance during inference under modality-complete conditions. However, a critical limitation emerges in practical deployments where various modality-missing scenarios frequently occur, leading to significantly degraded discriminative capabilities compared to idealized benchmarks. To address this challenge, the pioneering works (DENet [19] and TOP-ReID [9]) have conducted the initial investigation into pixel-level and token-level cross-modality reconstruction, aiming at handling the incompleteness of inference data. However, this paradigm inherently depends on fully observed multi-modality data for effective training. In real-world scenarios, due to data collection limitations, partially observed data streams will drastically compromise the training efficacy of such reconstruction-based approaches. This underscores the necessity of developing a robust multi-modality ReID model that works without requiring data completeness during both training and inference, ensuring its practical effectiveness in real-world applications.

Fortunately, with the advancements of Vision-Language foundation Models (VLMs) [20–25], their inherent cross-modal understanding capability has showcased transformative potentials in various multi-modality downstream tasks. Especially, by harnessing VLMs' open-world vision-text alignment, text-derived semantic features may effectively compensate for incomplete visual information, enabling robust solutions for modality-missing training and inference. Based on above insight, we specially deliver a flexible framework tailored for more realistic multi-modality retrieval scenario in this paper, dubbed as **Miss-ReID**, which is friendly to modality-missing scenarios without data-completeness assumption during both training and inference. As shown in Fig. 1 (b), the core of Miss-ReID lies in compensating for missing visual cues through vision-text knowledge transfer driven by VLMs. And the compensatory semantic-aligned textual features specifically excel at mitigating performance decline caused by partial visual modality absence.

Concretely, Miss-ReID mainly consists of three collaborative modules: Memory-based Heterogeneous Identity Prototype Representation (M-bHIPR) module, Modality-invariant Object Structure Modeling (M-iOSM) module, and Language-driven Missing Modality Completion (L-dMMC) module, as illustrated in Fig. 2. Firstly, M-bHIPR extracts diverse visual features from accessible modalities, and then builds modality-specific memory banks to store heterogeneous prototypes for each individual identity, ensuring the preservation of multi-modality characteristics. Afterwards, M-iOSM employs structure-aware query interactions to dynamically distill modality-invariant object structures from existing localized visual patches. By leveraging the *textual inversion* technique [26, 27], the extracted visual structural features are further reversed into pseudo-word tokens that encapsulate the identity-relevant structural semantics with L-dMMC module. Ultimately, the inverted tokens, integrated

with diverse learnable modality prompts, are embedded into crafted textual templates to form the personalized linguistic descriptions for diverse modalities. Benefiting from VLMs' inherent vision-text alignment capability, L-dMMC produces the textual embeddings to substitute the absent visual cues. These compensatory textual embeddings are further optimized through a memory-based contrastive constraint, thereby ensuring vision-text feature consistency. With the collaborations of M-bHIPR, M-iOSM and L-dMMC modules, our proposed Miss-ReID effectively compensates for the information absence caused by incomplete modalities, significantly improving retrieval performance under various modality-missing scenarios against state-of-the-art methods, as illustrated in Fig. 1 (c). At a glance, our major contributions are summarized as follows:

**(i)** To our knowledge, Miss-ReID is the first work to handle multi-modality ReID under more general modality-missing scenarios encountered during both training and inference. Our Miss-ReID allows the arbitrary modality-missing inputs, while preserving the multi-modality representation capacity, thereby propelling the advancement of multi-modality ReID toward real-world deployment.

**(ii)** Bolstered by the inherent vision-text reasoning capabilities of Vision-Language foundation Models (VLMs), Miss-ReID dynamically compensates for missing visual cues through semantic-aligned textual embeddings, and our intriguing findings highlight the potentials of developing VLMs within the realm of Multi-modality ReID encountering incomplete data streams.

**(iii)** Comprehensive experiments underscore our model's efficacy and superiority over state-of-the-art methods in various modality-missing retrieval scenarios, and our model demonstrates the lowest performance declines in mAP and Rank-1 accuracy compared to modality-complete evaluations on several benchmark datasets.

## 2   Related Work

**Multi-Modality Object ReID:**   Fueled by the complementary property from different modalities, multi-modality object ReID has drawn escalating research attention in recent years. For example, PFNet [28] is first proposed to progressively fuse features from diverse source modalities, enabling the extraction of discriminative multi-modality representations. EDITOR [11] is proposed to select object-centric tokens for filtering out irrelevant background information. DeMo [12] is designed to balance the decoupled hierarchical features using the mixture of experts, thereby enhancing feature robustness against variations in imaging quality across modalities. IDEA [10] is presented to construct text-enhanced multi-modality object ReID benchmarks, providing a structured caption generation pipeline. However, existing multi-modality studies typically assume the modality integrity during both training and inference, which strictly undermines the retrieval performance in the absence of partial modalities.

**Multi-Modality Learning with Missing Modality:**   Recently, several multi-modality learning methods [29–31, 18, 32–38] have prioritized improving the model's resilience against missing modalities. For instance, SMIL [30] utilizes the Bayesian Meta-Learning to simulate the latent features of missing modalities. ShaSpec [31] is proposed to explore shared-specific feature modeling framework to deal with missing modality in training and evaluation. Lee *et al.* [18] plug the learnable modality-missing-aware prompts into multi-modality transformers to identify different modality-missing inputs, thereby adapting the pre-trained transformer for various modality-missing tasks. Ke *et al.* [32] propose a training-free pipeline to address missing modality completion by leveraging the capabilities of Large Multimodal Models (LMMs). These advancements inspire us to work on completing multi-modality object representations under modality-missing retrieval scenarios.

## 3   Method

### 3.1   Preliminary

For simplicity and generality, we consider multi-modal ReID datasets comprising three modalities: RGB, Near Infrared (NIR) and Thermal Infrared (TIR) modalities. Formally, we define a modality-complete dataset as $\mathcal{D}_{com} = \{\mathcal{I}_{rgb}, \mathcal{I}_{nir}, \mathcal{I}_{tir}, \mathcal{Y}\}$, where $\mathcal{I}_m = \left\{I_m^i\right\}_{i=1}^{N_m}$ denotes the set of $N_m$ images in modality $m \in \{rgb, nir, tir\}$, and $\mathcal{Y} = \left\{y^i\right\}_{i=1}^{N}$ represents the identity labels for each paired triplet sample $(I_{rgb}^i, I_{nir}^i, I_{tir}^i)$. Under the modality-missing training paradigm proposed in

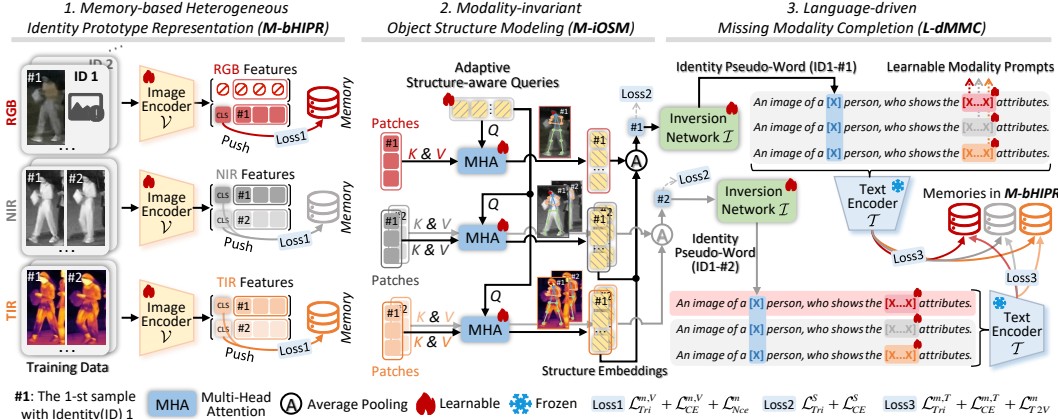

Figure 2: Pipeline of the proposed **Miss-ReID** under modality-missing training conditions.

this work, we construct a modality-incomplete dataset $\mathcal{D}_{mis} \subsetneq \mathcal{D}_{com}$, where certain modalities may be absent for specific samples by simulating random modality missing. Specifically, each modality $m$ is associated with a missing probability $\eta_m$, forming a probability tuple $\eta = (\eta_{rgb}, \eta_{nir}, \eta_{tir})$, *e.g.*, $\eta = (0.1, 0.1, 0.1)$ indicates a 10% chance of losing any single modality. For each triplet sample $(\tilde{I}_{rgb}^i, \tilde{I}_{nir}^i, \tilde{I}_{tir}^i, y^i)$ in $\mathcal{D}_{mis}$, we simulate modality absence by randomly setting $\tilde{I}_m^i$ to a zero-pixel tensor with probability $\eta_m$, or retaining the original $I_m^i$ with probability $1 - \eta_m$. Accordingly, the resulting modality-incomplete dataset $\mathcal{D}_{mis}$ enables us to explore robust multi-modality object ReID model under controlled modality absence.

## 3.2 Memory-based Heterogeneous Identity Prototype Representation (M-bHIPR)

As illustrated in the left of Fig. 2, M-bHIPR first extracts uni-modal visual features from accessible modalities. On that basis, it constructs an independent memory bank for each modality, storing heterogeneous identity prototypes (one per modality) to explicitly retain the characteristics of each modality. Technical implementations are elaborated as follows.

**Visual Feature Extraction:** Given the $i$-th triplet sample $(\tilde{I}_{rgb}^i, \tilde{I}_{nir}^i, \tilde{I}_{tir}^i)$ in modality-missing training dataset $\mathcal{D}_{mis}$, we capture its corresponding RGB, NIR or TIR visual features first, respectively, which is formulated as

$$f_m^i, F_m^i = \mathcal{V}(\tilde{I}_m^i | \theta_\mathcal{V}). \tag{1}$$

Here, $\mathcal{V}(* | \theta_\mathcal{V})$ expresses the siamese visual encoder derived from pre-trained VLMs (*e.g.*, CLIP) parameterized by $\theta_\mathcal{V}$. $f_m^i \in \mathbb{R}^{1 \times d}$ and $F_m^i \in \mathbb{R}^{N_l \times d}$ denote the produced global class embedding and local patch embeddings in modality $m \in \{rgb, nir, tir\}$, respectively. $N_l$ denotes the number of divided local patches, and $d$ is the embedding dimension.

**Prototype Initialization and Update Protocol:** To preserve and dynamically update the heterogeneous personal characteristics for each identity, we design a hierarchical memory architecture consisting of three modality-specific memory banks. Each bank stores identity-aware prototypes that encode the unique discriminative patterns within their respective modality (RGB, NIR, or TIR). The prototype initialization and update protocol is defined as follows:

For each identity $k$ in modality $m$, the initial prototype $p_{m,k}^{(0)}$ is computed as the feature centroid of all observed training samples belonging to identity $k$ in modality $m$, formulated as

$$p_{m,k}^{(0)} = \begin{cases} \frac{1}{|\mathcal{H}_m^k|} \sum_{f_m^j \in \mathcal{H}_m^k} f_m^j, & \text{if } \mathcal{H}_m^k \neq \emptyset, \\ \mathbf{0}, & \text{otherwise.} \end{cases} \tag{2}$$

Here, $\mathcal{H}_m^k$ indicates the set of whole available class embeddings with identity $k$ in modality $m$, $f_m^j$ denotes the $j$-th embedding contained in $\mathcal{H}_m^k$, and $|\mathcal{H}_m^k|$ counts the size of set. Notably, some

identities may lack samples in specific modalities under higher missing rate, *i.e.*, $\mathcal{H}_m^k = \emptyset$. Therefore, the zero-initialization serves as a placeholder mechanism when no modality-specific samples exist, maintaining the structural integrity of the memory bank.

After the $t$-th training epoch, each prototype stored in memory is updated using the corresponding class embeddings to integrate newly discriminative information while retaining historical knowledge in an Exponential Moving Average (EMA) way, written as

$$p_{m,k}^{(t)} = \alpha p_{m,k}^{(t-1)} + (1-\alpha) f_m^i, \ y^i = k. \tag{3}$$

Here, $\alpha \in (0, 1]$ is a momentum coefficient controlling the update smoothness, and is empirically set as 0.2. $f_m^i$ is the newly learned class embedding of the $i$-th image in modality $m$ with identity $k$ .

Critically, each modality-specific memory bank is updated using features derived exclusively from its corresponding modality. This design explicitly avoids cross-modality feature contamination during prototype refinement, thereby preserving the modality-specific object discriminative patterns within the respective representation space.

**Intra-modality Contrastive Optimization:**    On top of above established memory banks, the intra-modality contrastive loss function based on ClusterNCE [39] is designed to learn identity-invariant features within each modality. Specifically, for a given query $f_m^i$ from modality $m$, we compute its similarity with all identity prototypes stored in the corresponding memory bank, formulated as

$$\mathcal{L}_{Nce}^m = -\sum_{i=1}^N \log \frac{\sum_{k=1}^K \mathbb{1}_{[y^i=k]} \exp(f_m^i \cdot p_{m,k}/\tau)}{\sum_{k=1}^K \exp(f_m^i \cdot p_{m,k}/\tau)}, \tag{4}$$

where $y^i$ is the ground-truth identity label of the $i$-th sample tuple. $\mathbb{1}_{[y^i=k]}$ is an indicator function that equals 1 if $y^i = k$ (positive pair), and 0 otherwise (negative pair). $K$ denotes the number of all identities. $\tau$ is a temperature hyperparameter controlling the concentration level of the distribution, and is experimentally set to 0.05 here.

This optimization encourages our model to maximize feature-prototype alignment for positive pairs, while simultaneously minimizing cross-identity similarity for negative pairs. Such dual objectives cultivate the learning of identity-invariant features within each modality.

### 3.3   Modality-invariant Object Structure Modeling (M-iOSM)

The proposed M-iOSM hinges on an empirical observation that object structural configurations, *e.g.*, spatial relationships among body parts, or scene element compositions, typically exhibit remarkable consistency across visual modalities. *This suggests that structural patterns encode identity-specific information that is preserved regardless of the sensory modality.* This key insight motivates our approach: By distilling modality-invariant structural knowledge from available modalities, we aim to construct a robust bridge to facilitate subsequent feature completion for missing modalities within the L-dMMC module (Sec. 3.4). Its implementations are elaborated as follows.

**Adaptive Structure-aware Querying:**    As illustrated in the middle of Fig. 2, M-iOSM is designed to distill modality-invariant structural representations by adaptively querying object structural patterns from available modalities through a learnable query-based Multi-Head Attention mechanism (MHA).

In details, we initialize a set of $N_q$ learnable query vectors $\mathcal{Q} = [q_1, q_2, \cdots, q_{N_q}] \in \mathbb{R}^{N_q \times d}$ that serve as modality-shared "**Structural Probes**", which are expected to capture diverse structural patterns shared across modalities. Given $\mathcal{Q}$ and each input modality $m \in \{rgb, nir, tir\}$ with localized visual patches $F_m^i \in \mathbb{R}^{N_l \times d}$ derived by Eq. (1), we first project them into query, key and value spaces for each attention head $h$, respectively, written as

$$\mathbf{Q}^h = \mathrm{LN}(\mathcal{Q})\mathbf{W}_q^h, \quad \mathbf{K}_m^{i,h} = \mathrm{LN}(F_m^i)\mathbf{W}_k^h, \quad \mathbf{V}_m^{i,h} = \mathrm{LN}(F_m^i)\mathbf{W}_v^h, \tag{5}$$

where LN is the operation of Layer Normalization. $\mathbf{W}_q^h \in \mathbb{R}^{d \times d_H}$, $\mathbf{W}_k^h \in \mathbb{R}^{d \times d_H}$ and $\mathbf{W}_v^h \in \mathbb{R}^{d \times d_H}$ are three parameter-independent projection matrices. $d_H = d/N_H$ denotes the head size, and $N_H$ is the number of heads. After that, the adaptive structure-aware querying can be formulated as

$$\mathbf{W}_m^{i,h} = \mathrm{Softmax}\left((\mathbf{Q}^h \mathbf{K}_m^{i,h\top})/\sqrt{d_H}\right), \quad \mathbf{S}_m^{i,h} = \mathbf{W}_m^{i,h}\mathbf{V}_m^{i,h}, \quad \mathbf{S}_m^i = \mathrm{Cat}(\mathbf{S}_m^{i,1}, \cdots, \mathbf{S}_m^{i,N_H}). \tag{6}$$

Here, $\mathbf{W}_m^{i,h} \in \mathbb{R}^{N_q \times N_l}$, normalized by a Softmax function, denotes the weight matrix for perceiving the discriminative structural features from modality $m$ in the $h$-th attention head. $\mathbf{S}_m^{i,h} \in \mathbb{R}^{N_q \times d_H}$ represents the aggregated structural features. $\mathrm{Cat}(*)$ means concatenating features from all heads along channel dimension, resulting $\mathbf{S}_m^i \in \mathbb{R}^{N_q \times d}$ that represents the modality-invariant spatial structural features from modality $m$.

**Modality-invariant Structure Combination:** To obtain the information-complete and modality-invariant structural representations, we employ a straightforward yet effective approach to fuse features from all available modalities, expressed as

$$s_m^i = \frac{1}{N_q} \sum_{j=1}^{N_q} \mathbf{S}_m^i[j], \quad s^i = \frac{1}{M_i} \sum s_m^i, m \in \{rgb, nir, tir\}. \tag{7}$$

Here, $s_m^i$ denotes the intra-modality fused structural representation, and is assigned as zero tensor if modality $m$ is missing. $M_i$ denotes the number of available modalities in $i$-th triplet sample. Ultimately, the inter-modality fused representation $s^i$ is defined as the distilled structural feature that encapsulates the modality-invariant visual structural contexts of $i$-th triplet sample.

During training, the obtained structural representation $s^i$ is jointly supervised by the label smoothing cross-entropy loss to align structural features with identity labels, and triplet loss to enforce separation among different identities in the structural feature space. Simultaneously, the learnable query set $\mathcal{Q}$ is optimized to discover the most discriminative structural patterns through backpropagation, ensuring queries specialize in capturing identity-salient structural cues from multi-modality data.

### 3.4 Language-driven Missing Modality Completion (L-dMMC)

As depicted in the right of Fig. 2, the L-dMMC module is proposed to address the challenge of incomplete multi-modality data by leveraging linguistic priors to compensate for missing visual features, and the details are as follows.

**Inverted Identity Pseudo-word Generation:** To effectively and efficiently encapsulate identity-relevant structural semantics into textual tokens, we leverage a lightweight inversion network that transforms structural features into pseudo-word embeddings. Specifically, given the modality-invariant structural feature $s^i$ derived from Eq. (7), an inversion network maps $s^i$ into a continuous latent space, *i.e.*,

$$w_{inv}^i = \mathcal{I}(s^i | \theta_{\mathcal{I}}), \tag{8}$$

where $\mathcal{I}(* | \theta_{\mathcal{I}})$, implemented by a Multi-Layer Perceptron (MLP), denotes the inversion network parameterized by $\theta_{\mathcal{I}}$. And $\mathbf{w}_{inv}^i \in \mathbb{R}^{1 \times d}$ is defined as an inverted identity pseudo-word that effectively tells identity-specific visual structural contexts.

**Modality-specific Textual Prompting:** To complete VLM-compatible textual input, the inverted identity pseudo-word, combined with a set of learnable modality-specific prompts, are embed into a crafted textual template. For instance, template like "*An image of a [pseudo-word] person, who shows the [modality m] attributes.*" is employed to form the personalized linguistic descriptions tailored for specific modality. Wherein, **pseudo-word** $w_{inv}^i \in \mathbb{R}^{1 \times d}$ encodes identity-relevant structural semantics, while **prompts** $\mathcal{P}_m \in \mathbb{R}^{N_p \times d}$, $m \in \{rgb, nir, tir\}$, act as modality anchors, guiding the VLMs to interpret pseudo-word within the context of existing or missing modalities.

Afterwards, the tokenized textual template denoted as $\{t_1^i, t_2^i, \cdots, t_{N_t}^i\}$, combined with $w_{inv}^i$ and $\mathcal{P}_m$, is fed into the frozen text encoder to obtain textual embedding, formulated as

$$\tilde{f}_m^i = \mathcal{T}\left(\{t_1^i, t_2^i, \cdots, w_{inv}^i, \cdots, \mathcal{P}_m, \cdots, t_{N_t}^i\} | \theta_{\mathcal{T}}\right). \tag{9}$$

Here, $\mathcal{T}(* | \theta_{\mathcal{T}})$ denotes the text encoder derived from pre-trained VLMs (*e.g.*, CLIP) parameterized by $\theta_{\mathcal{T}}$. Therefore, $\tilde{f}_m^i \in \mathbb{R}^{1 \times d}$ is categorized as compensatory textual embedding when modality $m$ is missing, or as reconstructed textual embedding when modality $m$ is available.

**Memory-based Text-Vision Contrastive Optimization:** To ensure the compensatory or reconstructed textual embeddings align with the visual feature space and bridge the semantic gap between visual and textual modalities, we further propose the memory-based text-vision contrastive optimization strategy formulated as follows:

$$\mathcal{L}_{T2V}^m = -\sum_{i=1}^{N} \log \frac{\sum_{k=1}^{K} \mathbb{1}_{[y^i = k]} \exp(\tilde{f}_m^i \cdot p_{m,k} / \tau)}{\sum_{k=1}^{K} \exp(\tilde{f}_m^i \cdot p_{m,k} / \tau)}. \tag{10}$$

Here, $p_{m,k}$ denotes a visual prototype vector, which is derived from all observed training samples belonging to identity $k$ in modality $m$ by M-bHIPR module. Notably, the prototype $p_{m,k}$ is dynamically initialized and updated using $\tilde{f}_m^i$ associated with identity $k$ when identity $k$ lacks samples in modality $m$, with the aim to mitigate data scarcity issues. Conversely, $p_{m,k}$ remains unchanged if samples are available. This strategy ensures effective representation learning under varying data availability conditions. During training, each $\tilde{f}_m^i$ is encouraged to mimic the distribution of visual features corresponding to the same identity within modality $m$.

Above optimization enhances consistency between compensatory or reconstructed textual embeddings and their ground-truth visual features, thereby mitigating the absence of visual cues through the semantic-aligned textual representations under modality-missing scenarios.

### 3.5 Overall Objective Function

Beyond the contrastive learning objectives, *i.e.*, $\mathcal{L}_{Nce}^m$ and $\mathcal{L}_{T2V}^m$ defined in Eqs. (4) and (10), the visual, structural and textual features derived from M-bHIPR, M-iOSM and L-dMMC modules are also optimized through label smoothing cross-entropy losses (denoted as $\mathcal{L}_{CE}^{m,V}$, $\mathcal{L}_{CE}^S$, $\mathcal{L}_{CE}^{m,T}$) and triplet losses (denoted as $\mathcal{L}_{Tri}^{m,V}$, $\mathcal{L}_{Tri}^S$, $\mathcal{L}_{Tri}^{m,T}$), respectively, thereby facilitating their identity discrimination. Accordingly, the overall objective function of our Miss-ReID can be given by the following combination:

$$\mathcal{L}_{Total} = (\mathcal{L}_{Tri}^{m,V} + \mathcal{L}_{Tri}^S + \mathcal{L}_{Tri}^{m,T}) + \lambda_1(\mathcal{L}_{CE}^{m,V} + \mathcal{L}_{CE}^S + \mathcal{L}_{CE}^{m,T}) + \lambda_2(\mathcal{L}_{Nce}^m + \mathcal{L}_{T2V}^m). \quad (11)$$

Here, $m \in \{rgb, nir, tir\}$. $\lambda_1$ and $\lambda_2$ are hyper-parameters to balance the contributions of different terms. Notably, the terms $\mathcal{L}_{Tri}^{m,T}$, $\mathcal{L}_{CE}^{m,T}$ and $\mathcal{L}_{T2V}^m$ are incorporated into the optimizations after 20 epochs to stabilize the training.

## 4 Experiment

### 4.1 Datasets and Evaluation Protocols

**Datasets:** To evaluate our method under modality-missing scenarios, we conducted comprehensive experiments on multi-modality object ReID benchmarks (RGBNT201 [28] and RGBNT100 [40]) by introducing controlled data dropout during both training and inference phases. Specifically, for each modality-complete dataset, we randomly discard the partial data of each modality according to predefined tri-modality missing rates (*e.g.*, 10% for RGB, 30% for NIR, 50% for TIR) to simulate real-world sensor failures or data corruptions, generating modality-incomplete inputs for performance evaluation. **Evaluation Protocols:** Consistent with conventions in ReID community, two primary metrics— Cumulative Matching Characteristics at Rank-1 (**R-1** accuracy) and mean Average Precision(**mAP**)—are employed to assess model performance under seven inference scenarios: one modality-complete scenario (denoted as **RNT**, where all data modalities—**R**GB, **N**ear Infrared and **T**hermal Infrared—are fully available) and six modality-missing scenarios (denoted as R̲NT, RN̲T, RNT̲, R̲N̲T, R̲NT̲ and RN̲T̲). In these modality-missing scenarios, specific modalities in both query and gallery are omitted (*e.g.*, R̲NT excludes RGB images from the evaluation, and others omit specific modalities in analogous patterns). Notably, we also report **Mean mAP** and **Mean R-1** across the six modality-missing scenarios as the primary indicators for evaluating the model's holistic robustness against missing modalities.

### 4.2 Implementation Details

Our Miss-ReID is implemented using PyTorch libraries and runs on a single NVIDIA RTX A6000 GPU with 48GB VRAM. In line with prior works [9, 10], the pre-trained CLIP [20] is applied for the vision and text encoders. The model is trained in total of 50 epochs, with the L-dMMC module introduced after 20 epochs. We employ the Adam optimizer for training learnable modules, with a learning rate of $5e$-3 for modality prompts and $3.5e$-4 for others. The text encoder remains frozen throughout training. The number of structure-aware queries ($N_q$) is empirically set as 16 and 8 on RGBNT201 and RGBNT100, respectively. The length of modality prompts ($N_p$) is experimentally set as 4 per modality. $\lambda_1$ and $\lambda_2$ in Eq. (11) are both experimentally set to 0.1. Additionally, we summarize the overall training procedure in Algorithm 1 and illustrate the inference procedure in Fig. 4 under modality-missing conditions, which are available in **Appendices A.1** and **A.2**.

Table 1: The impacts of various components. We report the comparison results between different combinations (Model **B** – **F**) and the baseline (Model **A**) under both **modality-complete** and **-missing** training settings on RGBNT201. Here, '**Modality Complete**' represents learning the modality-complete data during **training**, and '$\eta = (0.1, 0.1, 0.1)$' denotes randomly abandoning 10% RGB images, 10% NIR images, and 10% TIR images during **training**. The evaluations are both conducted across six modality-missing scenarios, and mean mAP and R-1 are reported below.

| Index | Modules | | | Complexity | | Modality Complete | | $\eta = (0.1, 0.1, 0.1)$ | |
|---|---|---|---|---|---|---|---|---|---|
| | M-bHIPR | M-iOSM | L-dMMC | Params | FLOPs | Mean **mAP** | Mean **R-1** | Mean **mAP** | Mean **R-1** |
| **A** | ✗ | ✗ | ✗ | 86.4M | 34.3G | 48.9 | 50.4 | 46.4 | 47.0 |
| **B** | ✓ | ✗ | ✗ | 86.4M | 34.3G | 51.1(+2.2) | 51.4(+1.0) | 47.4(+1.0) | 48.0(+1.0) |
| **C** | ✗ | ✓ | ✗ | 86.4M | 34.3G | 50.2(+1.3) | 52.1(+1.7) | 46.9(+0.5) | 48.7(+1.7) |
| **D** | ✓ | ✓ | ✗ | 86.4M | 34.3G | 53.3(+4.4) | 54.1(+3.7) | 49.4(+3.0) | 49.8(+2.8) |
| **E** | ✗ | ✓ | ✓ | 89.6M | 43.6G | 52.1(+3.2) | 53.4(+3.0) | 47.4(+1.0) | 49.7(+2.7) |
| **F** | ✓ | ✓ | ✓ | 89.6M | 43.6G | 54.6(+5.7) | 55.7(+5.3) | 50.1(+3.7) | 51.3(+4.3) |

## 4.3 Ablation Study

As reported in Table 1, we conducted extensive ablation studies on RGBNT201 to evaluate the efficacy of individual components within Miss-ReID, under both modality-complete and modality-missing training settings.

**Baseline Settings:** The baseline method (Model **A**) relies solely on class embeddings derived from available modalities by visual encoder for retrieval, achieving 48.9% Mean mAP and 50.4% Mean R-1 accuracy under modality-complete training. When trained with 10% modality missingness ($\eta = (0.1, 0.1, 0.1)$), its performance drops to 46.4% Mean mAP and 47.0% Mean R-1, highlighting the challenge of modality incompleteness.

**Effectiveness of M-bHIPR:** Integrating the M-bHIPR module (Model **B**) improves modality-complete mAP by 2.2% (51.1%) and modality-missing mAP by 1.0% (47.4%). This indicates that M-bHIPR effectively aligns features with identity-related prototypes within each modality, enhancing discriminability while preserving modality-specific characteristics.

**Effectiveness of M-iOSM:** The proposed M-iOSM module (Model **C**) alone contributes 0.5% Mean mAP (46.9%) and 1.7% Mean R-1 (48.7%) improvements under modality-missing setting. When combined with M-bHIPR (Model **D**), performance surges to 49.4% Mean mAP (+3.0%) and 49.8% Mean R-1 (+2.8%). This indicates that M-iOSM fosters cross-modality interaction by dynamically mining similarities across modalities and modeling modality-invariant object structures, complementing M-bHIPR's intra-modality refinement. Notably, the M-bHIPR and M-iOSM modules operate without additional computational overhead during inference, preserving inference efficiency while enhancing multi-modality feature alignment.

**Effectiveness of L-dMMC:** The proposed L-dMMC module (Model **E**) alone yields 1.0% mAP (47.4%) and 2.7% Rank-1 (49.7%) gains under modality-missing training. Combined with all modules (Model **F**), it achieves the excellent performance: 54.6% Mean mAP (+5.7%) and 55.7% Mean R-1 (+5.3%) in modality-complete setting, with 50.1% Mean mAP (+3.7%) and 51.3% Mean R-1 (+4.3%) under missingness. L-dMMC leverages language priors to compensate missing modalities bolstered by the inherent vision-text reasoning capabilities of VLMs, enhancing robustness to various modality-missing scenarios. While parameters increase from 86.4M (Model **A**) to 89.6M (Model **F**), and FLOPs rise from 34.3G to 43.6G, the trade-off is justified by substantial accuracy improvements.

## 4.4 Comparisons with the State-of-the-art Methods

We benchmark our Miss-ReID against several state-of-the-art methods, including PCB [1], TOP-ReID [9], DeMo [12] and IDEA [10], under *modality-complete* training and *modality-missing* inference scenarios. Tables 2 summarizes the main results for multi-modality person ReID, evaluated on the RGBNT201 datasets. It is evident that our proposed Miss-ReID demonstrates significant robustness and superiority over SOTA methods in handling modality-missing challenges during inference. Specifically, Miss-ReID consistently outperforms the other methods in terms of both mAP and R-1 across the most modality-missing scenarios. Crucially, in the most challenging scenario

Table 2: Performance comparisons under **modality-missing** situations that only occur at the **inference** phase of multi-modality person ReID on RGBNT201. † denotes the model that is trained using both images and their corresponding text annotations. The best results are labeled with **boldface**. ↓x.x% and ↓x.x% highlight the lowest mAP and R-1 drop rates, respectively. '−' indicates that the metric is unpublished.

| Methods | RNT | | RNT | | RNT | | RNT | | RNT | | RNT | | RNT | | Mean | |
|---|---|---|---|---|---|---|---|---|---|---|---|---|---|---|---|---|
| | mAP | R-1 | mAP | R-1 | mAP | R-1 | mAP | R-1 | mAP | R-1 | mAP | R-1 | mAP | R-1 | mAP | R-1 |
| PCB [ECCV 2018] | 32.8 | 28.1 | 23.6 ↓28.0% | 24.2 ↓13.9% | 24.4 ↓25.6% | 25.1 ↓10.7% | 19.9 ↓39.3% | 14.7 ↓47.7% | 20.6 ↓37.2% | 23.6 ↓16.0% | 11.0 ↓66.5% | 6.8 ↓75.8% | 18.6 ↓43.3% | 14.4 ↓48.8% | 19.7 ↓39.9% | 18.1 ↓35.6% |
| TOP-ReID [AAAI 2024] | 72.3 | 76.6 | 54.4 ↓24.8% | 57.5 ↓24.9% | 64.3 ↓11.1% | 67.6 ↓11.7% | 51.9 ↓28.2% | 54.5 ↓28.9% | 35.3 ↓51.2% | 35.4 ↓53.8% | 26.2 ↓63.8% | 26.0 ↓66.1% | 34.1 ↓52.8% | 31.7 ↓58.6% | 44.4 ↓38.6% | 45.4 ↓40.7% |
| DeMo [AAAI 2025] | 79.0 | 82.3 | 63.3 ↓19.9% | 65.3 ↓20.7% | **72.6** ↓8.1% | **75.7** ↓8.0% | 56.2 ↓28.9% | 54.1 ↓34.3% | 45.6 ↓42.3% | 46.5 ↓43.5% | 26.3 ↓66.7% | 24.9 ↓69.7% | 40.3 ↓49.0% | 38.5 ↓53.2% | 50.7 ↓35.8% | 50.8 ↓38.3% |
| IDEA† [CVPR 2025] | **80.2** | **82.1** | 62.9 ↓21.6% | – ↓–% | 71.5 ↓10.8% | – ↓–% | 58.4 ↓27.2% | – ↓–% | 43.3 ↓46.0% | – ↓–% | 27.1 ↓66.2% | – ↓–% | 39.9 ↓50.2% | – ↓–% | 50.5 ↓37.0% | – ↓–% |
| **Miss-ReID** [**Ours**] | 76.9 | 78.9 | **66.6** ↓13.4% | **68.2** ↓13.6% | 72.4 ↓5.9% | 75.5 ↓4.3% | **63.2** ↓17.8% | **63.8** ↓19.1% | **47.2** ↓38.6% | **49.5** ↓37.3% | **34.5** ↓55.1% | **33.3** ↓57.8% | **43.9** ↓42.9% | **44.3** ↓43.9% | **54.6** ↓29.0% | **55.7** ↓29.4% |

where both RGB and TIR images are missing (**RNT**), Miss-ReID achieves 34.5% mAP and 33.3% R-1, which are significantly higher than those of the other methods. Notably, the colored boxes highlights Miss-ReID's dominance in minimizing performance decay under diverse missing-modality combinations. For instance, in **RNT** scenario, Miss-ReID experiences only 13.4% drop in mAP and 13.6% drop in R-1, which are the lowest drop rates among all compared methods. Moreover, Miss-ReID achieves new state-of-the-art average performance (54.6% mAP, 55.7% R-1), surpassing the second-best DeMo by 3.9% mAP and 4.9% R-1. The superior performance of Miss-ReID can be attributed to its textual feature completion tactic in handling modality-missing situations. Unlike other methods that may heavily rely on the presence of all modalities during both training and inference, Miss-ReID is designed to adaptively compensate missing modalities, maintaining high performance even when some modalities are unavailable. This makes Miss-ReID a more practical and reliable solution for real-world applications, where modality completeness cannot always be guaranteed. Moreover, the exhaustively comparative analysis with SOTA methods for modality-missing vehicle ReID on RGBNT100 dataset are provided in **Appendix A.3**.

### 4.5 Performance Analysis of Miss-ReID Under Varying Tri-modality Missing Rates

We conduct a comprehensive evaluation of our model's robustness to tri-modality missing rates $\eta = (\eta_{rgb}, \eta_{nir}, \eta_{tir})$ in multi-modality person ReID on RGBNT201. Table 3 reports performance under varying degrees of missing modalities during training, while assessing the model's behaviors in both modality-complete (**RNT**) and modality-missing (*e.g.*, **RNT**, **RNT**, etc.) scenarios during inference. As expected, increasing the missing rates inevitably degrades the model's performance across most evaluation scenarios, suggesting sensitivity to missing modalities during training. Nevertheless, the **Mean** row demonstrates that the model's average performance degrades gracefully as the missing rate increases, with mean mAP dropping from 54.6% in ideal case when $\eta = (0.0, 0.0, 0.0)$ to 47.3% in extreme case when $\eta = (0.5, 0.5, 0.5)$. This suggests that our model maintains reasonable robustness even under high missing rates, unlike some existing models (*e.g.*, PCB [1] with 19.7% mAP and TOP-ReID [9] with 44.4% mAP) trained exclusively on modality-complete data, which often struggle with missing modalities during inference.

### 4.6 Structure-aware Query Attention Region Visualization

As exhibited in Fig. 3, to intuitively showcase the efficacy of structure-aware queries in M-iOSM module, we specially provide visualizations about the most attentive region of each well-learned query vector across diverse scenarios. Each attentive region is highlighted by translucent mark (Q1-Q16) to facilitate intuitive interpretation, revealing two key insights. Firstly, within each modality, the distinct

Table 3: Performance comparisons of setting different **tri-modality missing rates** on RGBNT201. Each tuple $(\eta_{rgb}, \eta_{nir}, \eta_{tir})$ represents the proportion of randomly abandoned RGB, Near-Infrared, and Thermal-Infrared images during **training**.

| Tri-Modality Missing Rate $\eta$ | (0.0, 0.0, 0.0) | | (0.1, 0.1, 0.1) | | (0.3, 0.3, 0.3) | | (0.5, 0.5, 0.5) | | (0.1, 0.3, 0.5) | | (0.5, 0.3, 0.1) | |
|---|---|---|---|---|---|---|---|---|---|---|---|---|
| | mAP | R-1 | mAP | R-1 | mAP | R-1 | mAP | R-1 | mAP | R-1 | mAP | R-1 |
| **RNT** | 76.9 | 78.9 | 72.3 | 73.4 | 68.4 | 71.2 | 68.2 | 72.8 | 69.6 | 72.2 | 67.6 | 67.6 |
| **RNT** | 66.6 | 68.2 | 61.3 | 61.7 | 57.6 | 58.3 | 56.7 | 58.4 | 56.1 | 58.4 | 57.6 | 58.4 |
| **RNT** | 72.4 | 75.5 | 68.8 | 72.6 | 65.8 | 69.5 | 63.6 | 65.1 | 66.9 | 69.7 | 65.7 | 67.0 |
| **RNT** | 63.2 | 63.8 | 55.3 | 55.7 | 52.3 | 56.5 | 52.3 | 54.4 | 53.2 | 57.3 | 50.2 | 50.7 |
| **RNT** | 47.2 | 49.5 | 42.8 | 45.2 | 44.9 | 47.5 | 41.6 | 40.8 | 41.1 | 40.3 | 47.1 | 47.6 |
| **RNT** | 34.5 | 33.3 | 30.9 | 29.7 | 26.8 | 26.3 | 26.5 | 22.2 | 27.1 | 28.1 | 26.5 | 24.6 |
| **RNT** | 43.9 | 44.3 | 41.5 | 42.7 | 43.0 | 45.5 | 42.7 | 46.4 | 43.9 | 47.0 | 39.2 | 38.8 |
| **Mean** | **54.6** | **55.7** | **50.1** | **51.3** | **48.4** | **50.6** | **47.3** | **47.9** | **48.1** | **50.1** | **47.7** | **47.8** |

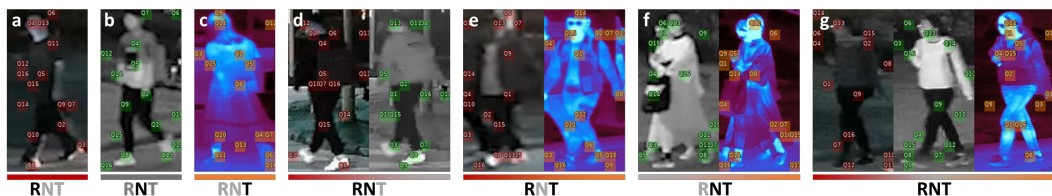

Figure 3: Visualizations of the attentive regions towards 16 well-learned structure-aware queries. We provide the results in 6 **modality-missing** cases (a-f: **RNT**, **RNT**, **RNT**, **RNT**, **RNT** and **RNT**), and a **modality-complete** scenario (g: **RNT**) sampled from RGBNT201 dataset.

structure-aware queries exhibit specialized focus on semantically meaningful body regions, *e.g.*, joint positions, accessory locations, etc. Critically, despite modality disparities, the object structure cues that excavated from different modalities with identical identity maintain contextual consistency, as displayed in Fig. 3 (d-g). These observations validate the robustness of our proposed M-iOSM module for probing the modality-invariant structure cues under real-world retrieval scenarios.

## 4.7 Further Evaluations and Analysis

To comprehensively evaluate the efficacy of our method, we additionally conduct extensive experiments, *encompassing more comparative evaluations, feature distribution visualizations, performance evaluations under 49 real-world inference scenarios, ranking lists, etc.* Please refer to the **Appendix A** for detailed results.

## 5 Conclusion

In this paper, a novel multi-modality object re-identification framework (Miss-ReID) has been proposed to furnish the robust retrieval under modality-missing conditions without requiring the data completeness during either training or inference. By harnessing the open-world vision-text alignment capabilities of Vision-Language foundation Models (VLMs), text-derived semantic features are nominated to effectively compensate for incomplete visual information, enabling robust solutions for modality-missing training and inference. Extensive experiments validate our model's efficacy and superiority over state-of-the-art methods across diverse modality-missing scenarios, advancing the practical deployment of multi-modality object re-identification.

**Limitations:** While the current Miss-ReID demonstrates the significant robustness across various modality-missing scenarios compared to state-of-the-art methods, further works are required to improve resilience in extreme cases (*e.g.*, complete modality collapse) and expanded modality integration (*e.g.*, event/LiDAR data, sketches, audio).

## Acknowledgments and Disclosure of Funding

I would like to extend my sincere gratitude to my doctoral supervisor, Prof. Qiang Zhang, for his patient guidance and valuable contributions to this work. This work was supported in part by China Postdoctoral Science Foundation under Grant 2023M742745, in part by the Postdoctoral Fellowship Program of China Postdoctoral Science Foundation (CPSF) under Grant GZB20230559, in part by Basic and Applied Basic Research Foundation of Guangdong Province under Grant 2023A1515110165, in part by the National Natural Science Foundation of China under Grant 61773301 and Grant 61803290, in part by the Fundamental Research Funds for the Central Universities under Grant No.ZYTS24022, and in part by the State Key Laboratory of Reliability and Intelligence of Electrical Equipment (Hebei University of Technology) under Grant EERI-KF2022005.

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

# A Appendix

In this supplementary material, we provide additional experimental evaluations, in-depth analyses and abundant visualizations to support our findings, and the structure is organized as follows:

## A.1 Training Procedure of Miss-ReID

We summarize the overall training procedure of our Miss-ReID under modality-missing conditions in Algorithm 1, where we explicitly consider three representative modalities, *i.e.*, RGB, Near Infrared (NIR) and Thermal Infrared (TIR), as the example.

---

**Algorithm 1:** Training procedure of **Miss-ReID**.

---

**Input:** Complete Multi-modality Object ReID Dateset: $\mathcal{D}_{com} = \{\mathcal{I}_{rgb}, \mathcal{I}_{nir}, \mathcal{I}_{tir}, \mathcal{Y}\}$;
Modality-missing Rate: $\eta = (\eta_{rgb}, \eta_{nir}, \eta_{tir})$; Learnable Structure-aware Queries: $\mathcal{Q}$;
Learnable Modality Prompts: $\mathcal{P}_m, m \in \{rgb, nir, tir\}$; Total Training Epoch: $E$.
**Output:** Robust Object ReID Model Under Modality-missing Conditions.

---

1 Build modality-incomplete dataset $\mathcal{D}_{mis} \subsetneqq \mathcal{D}_{com}$ by randomly dropping according to $\eta$;
2 Initialize memory bank for each modality by Eqs. (1) and (2);
3 **for** *epoch=1:E* **do**
4   *# Memory-based Heterogeneous Identity Prototype Representation.*
5   Extract multi-modality features of $\mathcal{D}_{mis}$ by image encoder $\mathcal{V}$ from pre-trained VLMs;
6   Update heterogeneous identity prototypes stored in memories in an EMA way by Eq. (3);
7   Calculate $\mathcal{L}_{Nce}^{m}$ using Eq. (4); Calculate $\mathcal{L}_{Tri}^{m,V}$ and $\mathcal{L}_{CE}^{m,V}$, $m \in \{rgb, nir, tir\}$.
8   *# Modality-invariant Object Structure Modeling.*
9   Employ $\mathcal{Q}$ to adaptively query modality-invariant structure embedding $s$ from available
   modalities using Eqs. (5–7); Calculate $\mathcal{L}_{Tri}^{S}$ and $\mathcal{L}_{CE}^{S}$.
10   **if** *epoch > 20* **then**
11    *# Language-driven Missing Modality Completion.*
12    Invert $s$ into identity pseduo-word $w_{inv}$ by inversion network $\mathcal{I}$ using Eq. (8);
13    Form three modality-specific text descriptions that incorporating $w_{inv}$ and $\mathcal{P}_m$;
14    Generate textual features by forzen text encoder $\mathcal{T}$ from pre-trained VLMs using Eq. (9);
15    Calculate $\mathcal{L}_{Tri}^{m,T}$ and $\mathcal{L}_{CE}^{m,T}$, $m \in \{rgb, nir, tir\}$;
16    Calculate $\mathcal{L}_{T2V}^{m}$ using Eq. (10), $m \in \{rgb, nir, tir\}$.
17   **else**
18    let $\mathcal{L}_{T2V}^{m} = 0$, $\mathcal{L}_{Tri}^{m,T} = 0$ and $\mathcal{L}_{CE}^{m,T} = 0$, $m \in \{rgb, nir, tir\}$.
19   **end**
20   Optimize network by minimizing Eq. (11).
21 **end**

---

## A.2 Textual Completion During Inference

Under various modality-missing inference scenarios, we substitute missing visual cues with semantically aligned text embeddings, thereby fostering robust multi-modality representations. Here, as illustrated in Fig. 4, we consider the scenario where RGB images are missing as an example to detail above process. Firstly, the visual features are captured from each available modality (*i.e.*, NIR and TIR) via VLMs' fine-tuned image encoder. Subsequently, the modality-invariant object structure cues are probed from two groups of local visual patches, and are further reversed into a

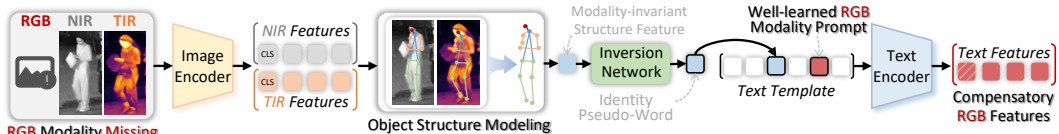

Figure 4: Illustration of our proposed textual feature completion tactic for missing modalities. Here, we take the absence of RGB image as an example.

Table 4: Performance comparisons under **modality-missing** situations only occurred at the **inference** phase of multi-modality vehicle ReID on RGBNT100. '–' indicates that the metric is unpublished.

| Methods | RNT | | RNT | | RNT | | RNT | | RNT | | RNT | | RNT | | Mean | |
|---|---|---|---|---|---|---|---|---|---|---|---|---|---|---|---|---|
| | mAP | R-1 | mAP | R-1 | mAP | R-1 | mAP | R-1 | mAP | R-1 | mAP | R-1 | mAP | R-1 | mAP | R-1 |
| DENet [arXiv 2023] | 68.1 | 89.2 | – | – | 62.0 | 85.5 | 56.0 | 80.9 | – | – | – | – | 50.1 | 74.2 | – | – |
| | | | – | – | ↓9.0% | ↓4.1% | ↓17.8% | ↓9.3% | – | – | – | – | ↓26.4% | ↓16.8% | – | – |
| TOP-ReID [AAAI 2024] | 81.2 | 96.4 | 70.6 | 90.6 | 77.9 | 94.5 | 64.0 | 81.5 | 42.5 | 69.3 | 45.9 | 65.4 | 55.4 | 77.8 | 59.4 | 79.9 |
| | | | ↓13.1% | ↓6.0% | ↓4.1% | ↓2.0% | ↓21.2% | ↓15.5% | ↓47.7% | ↓28.1% | ↓43.5% | ↓32.2% | ↓31.8% | ↓19.3% | ↓26.8% | ↓17.1% |
| DeMo [AAAI 2025] | **86.2** | **97.6** | **81.0** | 94.5 | **84.1** | **96.5** | **71.1** | 87.6 | **50.2** | 73.7 | **59.6** | 78.1 | **66.3** | 82.8 | **68.7** | 85.5 |
| | | | ↓6.0% | ↓3.2% | ↓2.4% | ↓1.1% | ↓17.5% | ↓10.2% | ↓41.8% | ↓24.5% | ↓30.9% | ↓20.0% | ↓23.1% | ↓15.2% | ↓20.3% | ↓12.4% |
| **Miss-ReID** [**Ours**] | 82.5 | 96.7 | 77.3 | **94.9** | 81.9 | 96.3 | 70.3 | **89.7** | 47.1 | **78.0** | 57.7 | **78.2** | 65.6 | **86.8** | 66.6 | **87.3** |
| | | | ↓6.3% | ↓1.9% | ↓0.7% | ↓0.4% | ↓14.8% | ↓7.2% | ↓42.9% | ↓19.3% | ↓30.1% | ↓19.1% | ↓20.5% | ↓10.2% | ↓19.3% | ↓9.7% |

pseudo-word that encapsulates the identity-related visual structural contexts. Ultimately, integrated with well-learned RGB modality prompts, inverted token are inserted into text template to form the input for VLMs' frozen text encoder. Benefiting from VLMs' inherent image-text alignment capability, the resulting textual features serve as the compensatory features for missing RGB modality. Therefore, by concatenating the existing NIR and TIR visual features with the compensatory RGB textual features, we enable robust multi-modality object re-identification under modality-missing inference condition.

## A.3 Comparative Analysis with SOTA Methods for Modality-missing Vehicle ReID

We also benchmark our Miss-ReID against several state-of-the-art methods, including DENet [19], TOP-ReID [9] and DeMo [12], for multi-modality vehicle ReID under *modality-complete* training and *modality-missing* inference scenarios. Notably, the textual template in L-dMMC module is crafted as "*An image of a [pseudo-word] vehicle, which shows the [modality m] attributes*" here. Tables 4 reports the main results evaluated on the RGBNT201 datasets. Obviously, th proposed Miss-ReID demonstrates superior performance and robustness over SOTA methods across diverse modality-missing combinations. To be specific, Miss-ReID exhibits minimal performance degradation in critical scenarios: under NIR-missing case (**RNT**), its R-1 accuracy drops by only 0.4% (96.3% vs. 96.7%), outperforming DeMo's 1.1% decline, TOP-ReID's 2.0% drop and DENet's 4.1% reduction; under TIR-missing case (**RNT**), it maintains an R-1 accuracy decrease of 7.2% (89.7% vs. 96.7%), surpassing DeMo (10.2%) and TOP-ReID (15.5%) by 30% to 50%; even in the most challenging case where both RGB and TIR images are missing (**RNT**), it achieves 78.2% R-1 accuracy (19.1% drop), exceeding DeMo (78.1% with 20.0% drop) and TOP-ReID (65.4% with 32.2% drop). Across all scenarios, Miss-ReID achieves 66.6% mAP and 87.3% R-1 with degradation rates of 19.3% (mAP) and 9.7% (R-1), outperforming DeMo (20.3%/12.4%) and TOP-ReID (26.8%/17.1%) in generalization under modality uncertainty. Compared to DeMo, Miss-ReID shows smaller degradation in 5/6 modality-missing scenarios in terms of R-1 accuracy, while outperforming TOP-ReID by reducing average degradation by 7.2% (mAP) and 7.4% (R-1). These results validate the effectiveness of our language-driven missing modality completion approach, which enables Miss-ReID to serve as a robust multi-modality vehicle ReID solution for real-world deployments where partial modality failures frequently occur.

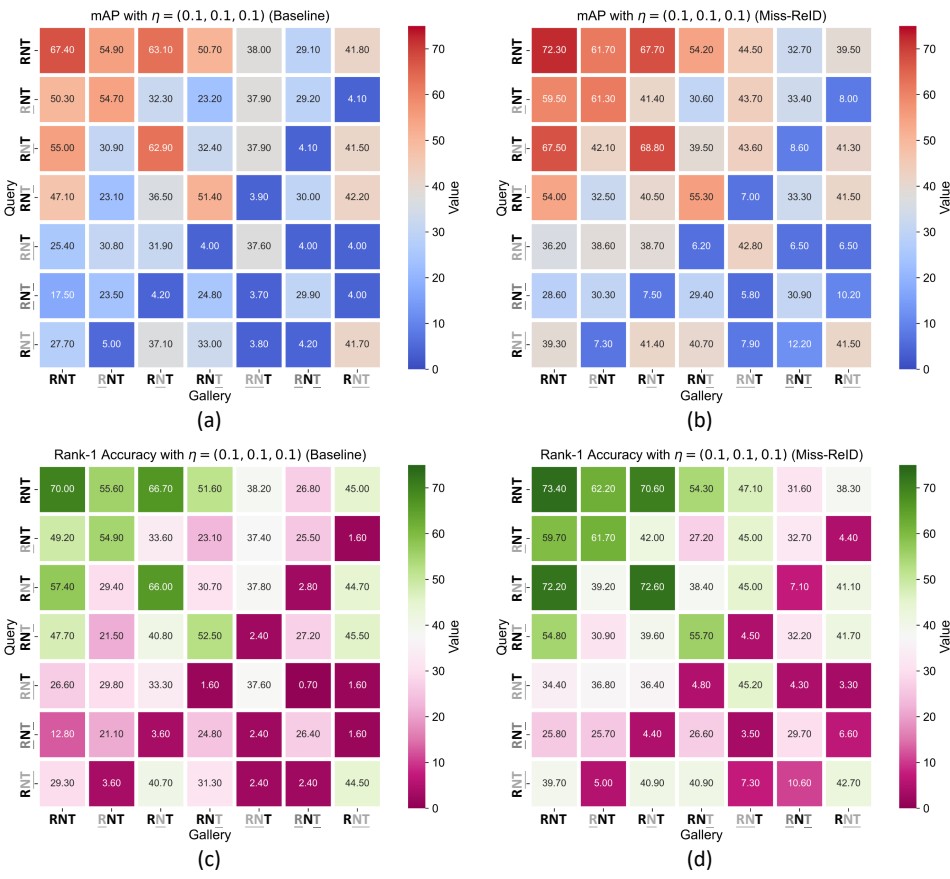

Figure 5: Matrix visualizations of the retrieval performances, *i.e.*, (a-b) Mean mAP, and (c-d) Mean Rank-1, on the RGBNT201 dataset. Here, the baseline model and our Miss-ReID are both trained using **modality-missing** data with $\eta = (0.1, 0.1, 0.1)$, and are evaluated under 1 **modality-complete** inference scenario ("**RNT**-to-**RNT**") and else 48 more general **modality-missing** inference scenarios.

## A.4 Retrieval Performances under 49 Real-world Scenarios in Tri-modality ReID

To comprehensively evaluate the robustness of our Miss-ReID towards incomplete modalities, we consider 49 more general "*Query-to-Gallery*" retrieval scenarios, encompassing 1 modality-complete case ("**RNT**-to-**RNT**") and else 48 modality-missing cases. As illustrated in Fig. 5, we visualize the performance comparisons between the baseline model (a, c) and our Miss-ReID (b, d). It's evident that our Miss-ReID outperforms the baseline model in both modality-complete and -missing scenarios, as evidenced by the higher Mean mAP and Mean Rank-1 accuracy. Specifically, in the modality-complete case, our model demonstrates superior performance, indicating its effectiveness in leveraging all available modalities for accurate retrieval. Furthermore, even in else modality-missing cases, which represent more challenging and realistic scenarios, our model consistently achieves better retrieval results compared to the baseline. While it is true that in certain extreme modality-missing scenarios (*e.g.*, "**R̲N̲T**-to-**R̲N̲T**" and "**RN̲T̲**-to-**RN̲T̲**"), our model's performance is somewhat limited, it still maintains an advantage over the baseline model. This highlights the robustness of our approach but also points to an area for future improvement. Enhancing the model's ability to handle extreme modality missing will be a key focus in our future work, further boosting retrieval performances in these challenging cross-modality conditions.

## A.5 Feature Distribution Visualization

As shown in Fig. 6, to intuitively witness the efficacy of the compensatory textual features derived from L-dMMC module, we visualize the distributions of three types of discriminative features under challenging modality-missing case (**R̲N̲T**), where both RGB and TIR images are unavailable. From

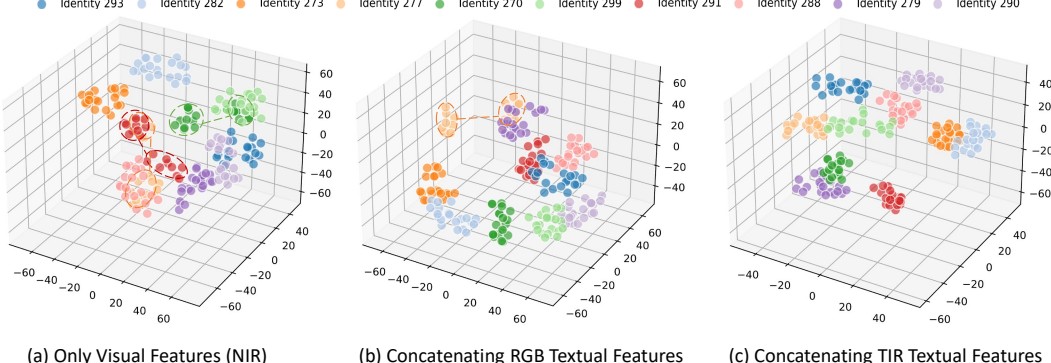

(a) Only Visual Features (NIR)  (b) Concatenating RGB Textual Features  (c) Concatenating TIR Textual Features

Figure 6: The feature distributions of 10 identities randomly sampled from the RGBNT201 dataset by using t-SNE [41]. Here, we take the challenging case (**RNT**) that both RGB and TIR images are unavailable as an example. (a) The vision-only features derived from available NIR images. (b) The fused features that concatenating the compensatory RGB textual features on (a). (c) The fused features that further concatenating the compensatory TIR textual features on (b). Different colors refer to different identities.

Fig. 6 (a) to Fig. 6 (c), the features of challenging samples (specifically, IDs 270, 277, and 291) become increasingly compact, while the separation between different identities (IDs) widens. These visualizations fully substantiate that progressively incorporating the compensatory RGB and TIR textual features into limited visual features significantly enhances the feature discrimination and robustness towards modality-missing cases.

## A.6  Retrieval Results Under Both Modality-complete and -missing Situations

As shown in Fig. 7, we compare the ranking lists generated by (a) the baseline model and (b) our proposed Miss-ReID, under both modality-complete and modality-missing inference scenarios. The baseline model demonstrates limited performance, yielding a high number of incorrect matches, particularly in the most challenging modality-missing cases (**RNT**, **RNT**, and **RNT**). In contrast, our Miss-ReID achieves superior performance, with significantly fewer incorrect matches and more accurate results. These findings intuitively validate the effectiveness of our approach in compensating for missing modalities using textual features.

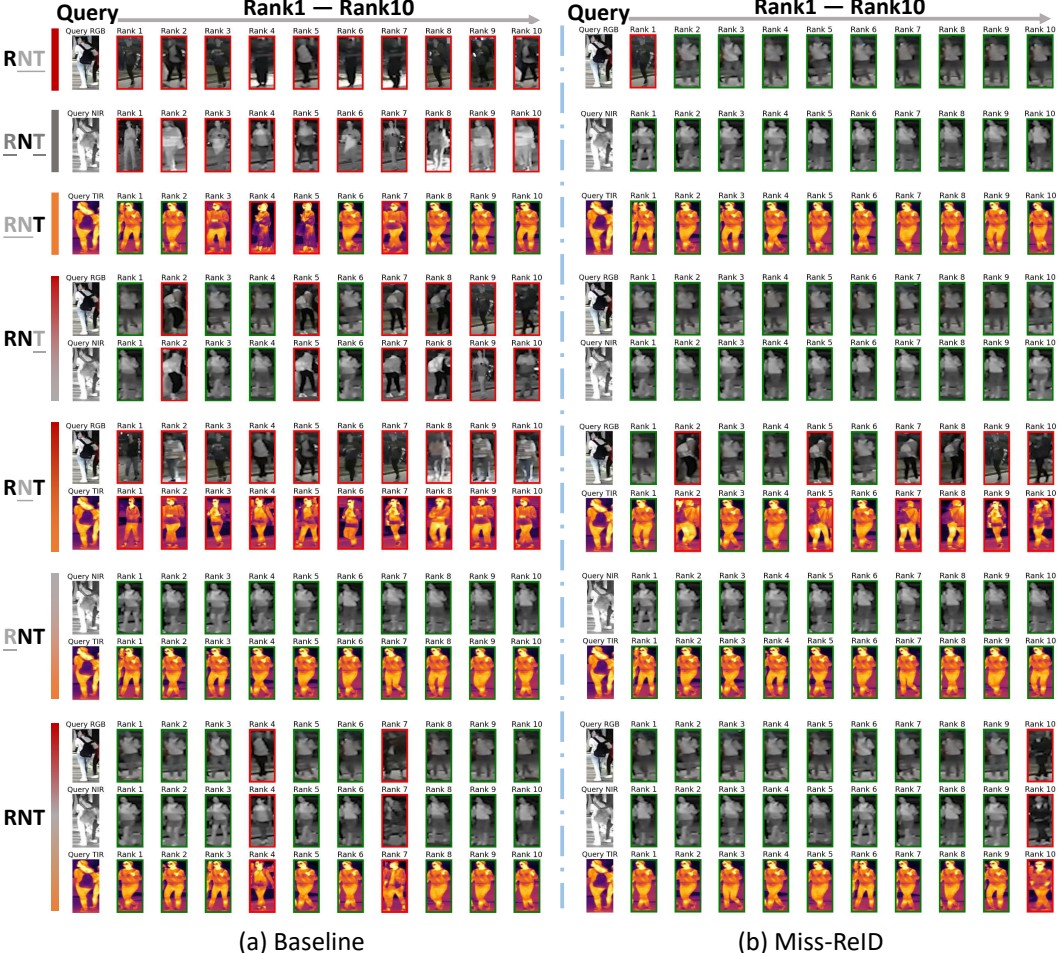

Figure 7: Ranking list comparison between (a) Baseline and (b) Our Miss-ReID under one **modality-complete** retrieval scenario and 7 **modality-missing** retrieval scenarios. The green box denotes the correct match, whereas the red box signifies the incorrect match.

