# OpenReview forum: "Miss-ReID: Delivering Robust Multi-Modality Object Re-Identification Despite Missing Modalities"
_NeurIPS.cc/2025/Conference — NeurIPS 2025 poster_

### Official Review · Reviewer_uuXE · 2025-06-29

**Clarity:** 3
**Significance:** 4
**Originality:** 3
**Rating:** 5
**Confidence:** 5

**Summary:**

The paper presents Miss-ReID, a flexible framework for robust multi-modality object re-identification (ReID) under modality-missing conditions. It is the first to support both modality-missing training and inference, leveraging Vision-Language foundation Models (VLMs) for vision-text knowledge transfer to compensate for missing visual cues. The framework comprises three modules: Memory-based Heterogeneous Identity Prototype Representation (M-bHIPR) to store modality-specific prototypes, Modality-invariant Object Structure Modeling (M-iOSM) to distill structural features, and Language-driven Missing Modality Completion (L-dMMC) to generate compensatory textual embeddings via inverted pseudo-word tokens and modality prompts. Extensive experiments on RGBNT201 and RGBNT100 show that Miss-ReID outperforms state-of-the-art methods in various modality-missing scenarios, demonstrating the lowest performance declines and advancing multi-modality ReID for real-world applications.

**Questions:**

1. The framework leverages VLMs for modality compensation, which might incur higher computational costs. Could the authors provide inference time comparisons between Miss-ReID and state-of-the-art methods under modality-missing scenarios to clarify its practical efficiency?
2. The M-bHIPR module maintains modality-specific memory banks, but when the modality missing rate is high, does the prototype update strategy in Eq. (3) ensure effective preservation of identity characteristics? Are there additional experiments to verify its robustness under low-sample conditions?

**Ethical Concerns:**

["NO or VERY MINOR ethics concerns only"]

**Final Justification:**

I am pleased to raise my rating to 5, and support acceptance of this paper.

**Limitations:**

Yes. The authors have explicitly discussed both technical limitations further works are required to improve resilience in extreme cases (e.g., complete modality collapse) and expanded modality integration (e.g., event/LiDAR data, sketches, audio).

**Paper Formatting Concerns:**

This paper is well-formatted and compliant with NeurIPS 2025 guidelines, requiring no significant changes.

**Quality:**

4

**Strengths And Weaknesses:**

Strengths:

1. It is the first work to handle multi-modality ReID under general modality-missing scenarios during both training and inference, enabling arbitrary modality-missing inputs and promoting real-world deployment of multi-modality ReID.
2. By leveraging the inherent vision-text reasoning capabilities of Vision-Language foundation Models (VLMs), it dynamically compensates for missing visual cues through semantic-aligned textual embeddings, showing great potential in addressing incomplete data streams in Multi-modality ReID.
3. The framework consists of three collaborative modules (M-bHIPR, M-iOSM, and L-dMMC), which work together to extract features, model structures, and complete missing modalities, ensuring the preservation of multi-modality characteristics and effective compensation for information absence.
4. Extensive experiments on benchmark datasets demonstrate its superiority over state-of-the-art methods in various modality-missing scenarios, with the lowest performance declines in mAP and Rank-1 accuracy, verifying its efficacy and robustness.

Weaknesses:

1. It needs further improvement in resilience for extreme cases like complete modality collapse, which limits its performance in some harsh conditions.
2. The integration of VLMs may lead to higher computational requirements, potentially posing challenges for low-resource or real-time deployment scenarios, there is a certain increase in parameters and FLOPs.

---

> ### Author Rebuttal · Authors · 2025-07-29
>
> We are truly grateful for your constructive and insightful feedback. Your recognition of our work means a great deal, and we deeply appreciate your support. Below is our detailed, point-by-point response to all the identified Questions (Q) and Weaknesses (W).
>
> ---
>
> **Q1/W2: Model complexity and inference speed.**
>
> We appreciate your valuable concerns about computational complexity and inference speed, which are critical for real-world deployment. As suggested, we provide detailed comparisons about model complexity and inference speed between Miss-ReID and SOTA methods.
> The table below reports key metrics: trainable parameters, FLOPs, and FPS across both modality-complete and modality-missing scenarios. All evaluations were conducted on an NVIDIA RTX A6000 GPU.
>
> |Methods|Backbone|Params (Trainable)| FLOPs|FPS (No Miss)|FPS (1 Miss)|FPS (2 Miss)|Mean mAP|
> |:------------|:-----:|:-----:|:-----:|:-----:|:-----:|:-----:|:-----:|
> | DeMo (25'AAAI)   |CLIP(ViT-B/16)  |98.8 M| 35.1 G| 28.7 | 47.7| 97.8 |50.7|
> | IDEA (25'CVPR)  |CLIP(ViT-B/16)  |91.7 M|43.6 G|  15.9 | 24.1  | 46.3 |50.5|
> | Miss-ReID (Ours) |CLIP(ViT-B/16)  |89.6 M|43.6 G|37.5 | 34.9 | 35.1   |54.6|
>
> - _**Inference speed**_:  Our Miss-ReID strikes **a favorable balance between performance and inference speed** across both modality-complete and -missing scenarios. Specifically, it outperforms SOTA methods (DeMo, IDEA) in Mean mAP while meeting real-time requirements (>30 FPS), making it suitable for practical systems like surveillance.
> In modality-complete setting, our model achieves 37.5 FPS, significantly exceeding IDEA (15.9 FPS) and DeMo (28.7 FPS). This efficiency advantage thanks to its selective VLM activation only for missing modalities, balancing performance and speed.  In modality-missing scenarios (especially with 2 modalities missing), Miss-ReID maintains stable real-time speed (35.1 FPS) while achieving the best mAP. In contrast, DeMo and IDEA fail to match our performance despite higher speeds.
>
> - _**Computational costs**_:  While Miss-ReID utilizes similar training FLOPs (43.6 G) to IDEA (CVPR'25), our model has fewer trainable parameters (89.6 M) than others (91.7 M and 98.8 M), thereby reducing training memory overhead.
>
> - _**Lightweight variant**_: We plan to develop a lightweight variant of Miss-ReID for edge devices, focusing on three optimizations: 1) Prompt Distillation via KL divergence to compress text encoder layers (50% reduction); 2) Structured Pruning to remove redundant attention heads (20% fewer parameters); 3) Replacing ViT-B with MobileViT-XS, slashing backbone FLOPs by 60%. These aim to deliver edge-ready performance while preserving core compensation capabilities.
>
> We will add these details in the revision to clarify practical efficiency.
>
> ---
>
> **Q2/W1: Robustness in some harsh conditions.**
>
> We appreciate these critical observations on robustness under harsh conditions. Regarding extreme modality loss (e.g., high missing rates), we supplement experimental results under varying tri-modality missing rates ($ \eta = (\eta_{rgb}, \eta_{nir}, \eta_{tir}) $, representing the proportion of missing samples per modality), as reported in Table below.
>
> | Missing Rates|$ \eta $ =  (0.5, 0.5, 0.5) | $ \eta $ =  (0.7, 0.7, 0.7) | $ \eta $ =  (0.9, 0.9, 0.9) |
> |:------------:|:-----:|:-----:|:-----:|
> | Mean mAP |  47.3  |  38.6 | 12.9|
> | Mean R-1   |  47.9  |  36.7 | 9.7|
>
> These results indicate that, under moderate high rates, _i.e._, $ \eta $ =  (0.7, 0.7, 0.7), our model retains reasonable performance (mAP=38.6), demonstrating resilience by leveraging memory banks and textual compensation.
> As expected, at extreme rates, _i.e._, $ \eta $ =  (0.9, 0.9, 0.9), performance drops significantly, as limited remaining samples weaken prototype reliability in M-bHIPR and reduce structural cues for L-dMMC, which is consistent with the challenge of near-complete modality collapse.
>
> This decline can be attributed to two factors. **First**, the M-bHIPR module’s prototype update (Eq. 3), which currently averages features from available samples, struggles when data is scarce, _e.g._, $ \eta $ =  (0.9, 0.9, 0.9).
> **Second**, insufficient training data undermines the model’s ability to learn robust features.
> This highlights the need to enhance model robustness under low-sample conditions.
>
> **Our future improvements will include**: **1)** Dynamically weighting prototypes in M-bHIPR (_e.g._, prioritizing high-confidence samples and using knowledge distillation to retain historical reliable prototypes);
> **2)** Strengthening L-dMMC with cross-modality attention to extract richer semantics from sparse remaining cues;
> **3)** Adding adversarial training for extreme cases, _e.g._, $ \eta $ =  (0.9, 0.9, 0.9), to boost model generalization.
>
> We will include these analyses and plans in the revision, along with comparisons to adapted baselines under high missing rates.
>
> ---
>
> Thank you sincerely for your valuable insights, which have greatly refined our work. We hope our responses address the raised concerns thoroughly, and we’d be honored by your support for our submission. Thank you for your time and expertise.

---

### Official Review · Reviewer_zX11 · 2025-07-01

**Clarity:** 3
**Significance:** 3
**Originality:** 3
**Rating:** 4
**Confidence:** 5

**Summary:**

This paper proposes Miss-ReID, a novel framework for multi-modality object re-identification (ReID) that is robust to missing modalities during both training and inference. Unlike existing methods that rely on complete multi-modal inputs, Miss-ReID introduces a three-part architecture: (1) Memory-based Heterogeneous Identity Prototype Representation (M-bHIPR) to preserve modality-specific prototypes, (2) Modality-invariant Object Structure Modeling (M-iOSM) to extract shared structural features across modalities, and (3) Language-driven Missing Modality Completion (L-dMMC), which uses Vision-Language Models (VLMs) to convert visual structure into compensatory textual features. The model dynamically constructs personalized textual prompts to compensate for absent modalities, leveraging CLIP’s cross-modal alignment. Experiments on the RGBNT201 and RGBNT100 datasets demonstrate that this work reaches a satisfying performance.

**Questions:**

1）The L-dMMC module uses fixed modality prompts (e.g., “[modality m] attributes”), but it does not explain how these prompts adapt to different missing modality scenarios. For instance, when the RGB modality is missing, should the prompt provide more specific semantic guidance (e.g., “color characteristics under visible light”)? It is recommended to design ablation experiments to compare the performance of fixed prompts versus dynamically generated prompts, and to elaborate on the theoretical rationale behind the prompt design.

**Ethical Concerns:**

["NO or VERY MINOR ethics concerns only"]

**Final Justification:**

Thank you to the authors for their response. Some of my questions have been resolved. Therefore, I will maintain my original score.

**Limitations:**

Please refer to Weaknesses.

**Quality:**

3

**Strengths And Weaknesses:**

Strengths:

1）This paper is the first to systematically address the practical challenge of modality-missing scenarios occurring during both training and inference in multi-modality ReID. By introducing a language-driven modality completion mechanism and leveraging the semantic alignment capabilities of vision-language foundation models, the proposed method exhibits strong generalizability and adaptability, representing a significant and original contribution to the field.

2）The proposed three-module architecture—M-bHIPR, M-iOSM, and L-dMMC—is conceptually coherent and functionally complementary, effectively addressing modality-specific feature preservation, modality-invariant structure modeling, and semantic compensation. The extensive experimental validation, including performance comparisons under diverse missing-modality conditions and detailed ablation studies, demonstrates both the robustness and practical applicability of the framework.

Weaknesses:

1）The core of the model relies on the vision-text alignment capabilities of VLMs (e.g., CLIP) to compensate for missing modalities; however, the paper does not investigate the impact of different VLMs on performance. For example, when replacing CLIP with VLMs of varying parameter scales or pretraining data (such as BLIP-2 or InstructBLIP), the semantic representation capacity of the generated textual features may differ significantly, potentially leading to instability across VLM variants. Furthermore, both the visual and textual encoders of the VLMs are kept frozen, and the adaptation is performed solely through pseudo-word generation and modality-specific prompts. This "lightweight" tuning strategy may not fully exploit the capabilities of different VLMs, thereby limiting the model's generalizability in diverse application scenarios.

2）The model employs structure-aware querying (M-iOSM) to extract modality-invariant structural features, but it relies solely on the spatial relationships of local patches (e.g., spatial configurations of body parts) and overlooks fine-grained semantic cues such as clothing textures for pedestrians or component details for vehicles. Moreover, the inversion process from structural features to pseudo-words (L-dMMC) uses a simple MLP mapping, which lacks the capacity to explicitly model semantic attributes like “color” or “material.” As a result, the generated compensatory textual features may suffer from semantic deficiency when representing complex identities.

---

> ### Author Rebuttal · Authors · 2025-07-29
>
> We want to express our gratitude for your valuable insights.
> Your recognition of our work is deeply valued, and we’re truly grateful for your support of this research!
> Below is our detailed, point-by-point response to all the identified Weaknesses (W) and Questions (Q).
>
> ---
>
> **W1:  Some issues about using VLMs.**
>
> _**a. CLIP vs. BLIP-2**_:
> We appreciate your insightful suggestion regarding the impact of different VLMs.
> Following your advice, we evaluated various VLMs as backbones (**with only text encoders frozen during training**), and the results are presented in the table below:
>
> |Backbone| Image Encoder| Text Encoder| Params (Trainable)| Params (Total)| Mean mAP|Mean R-1|
> |:------------:|:-----:|:-----:|:-----:|:-----:|:-----:|:-----:|
> | CLIP    |  ViT-B/16  | Transformer| 89.6M   |   153.0 M   |  54.6  | 55.7|
> | BLIP-2 |  ViT-L/14  | OPT-2.7B    | 310.6M |      3.1B     |  55.4  | 56.4 |
>
> The table shows that while BLIP-2 (ViT-L/14 + OPT-2.7B) has 3.4× trainable parameters (310.6M vs. 89.6M) and 20× total parameters compared to CLIP, its performance gains (mAP +0.8%, R-1 +0.7%) are marginal.
> This suggests stronger VLMs don’t guarantee proportional improvements in specific task, which may due to the following two reasons.
>
> - **1)** BLIP-2’s massive scale (3.1B parameters) demands larger training data to unleash its potential, and relatively small-scale ReID datasets (e.g., RGBNT201 with 4,787 aligned RGB, NIR and TIR images) cannot provide this.
>
> - **2)** BLIP-2 excels at generative tasks (e.g., image captioning), while CLIP’s contrastive pretraining aligns better with cross-modality retrieval objectives, offering a more efficient feature space for ReID.
>
> Thus, despite BLIP-2's stronger general capabilities, CLIP remains a more optimal choice for our task due to its better alignment with retrieval objectives and lower computational burden. We will elaborate on these in the revision.
>
>
> _**b. Fine-tuning strategy**_:
> We sincerely apologize for the ambiguity in our previous description regarding the fine-tuning strategy.
> Our Miss-ReID **only freezes the textual encoder** to preserve VLMs' rich semantic knowledge **while fine-tuning the visual encoder** through memory-based contrastive learning, enabling it to learn identity-specific features aligned with textual semantics.
> Therefore, by retaining the pre-trained textual encoder’s robust semantic foundation and adapting the visual encoder to **ReID-specific cues**, this strategy balances lightweight tuning with effective exploitation of VLMs, boosting generalizability across scenarios.
> More detailed training procedure is provided in Algorithm 1 in the submission.
>
> ---
>
> **W2:  Some issues about ''Structure Modeling'' and ''Inversion Network Design''.**
>
> We sincerely appreciate your insightful comments regarding ''Structure Modeling'' and ''Inversion Network Design'', and our detailed responses are as follows.
>
> _**a. Fine-grained cues exploration during structure modeling.**_
>
> The M-iOSM module indeed prioritizes spatial relationships as foundational modality-invariant cues. Notably, their modality-invariant nature is critical for cross-modality compensation, as they provide a stable structural foundation that, when combined with modality prompts, enables effective supplementation of identity-specific information for missing modalities.
> Additionally, our ablation studies have validated the significance of these structural features.
> **More importantly, we fully agree that integrating fine-grained cues will further enhance the richness of compensatory features.** In the future, as you suggested, the improvements will be as follows:
>
> - **1)** We plan to introduce **fine-grained attribute heads** (e.g., texture, material) in M-iOSM to explicitly model these cues while preserving structural invariance.
> - **2)** We plan to adopt **multi-scale patch attention**, where smaller patches focus on local details like fabric textures, complementing the existing spatial relationship modeling.
>
> _**b. MLP-based inversion network design.**_
>
> Textual inversion, as a prevalent approach in recent multi-modal research, aims to discover new pseudo-words within the word-embedding space. **Following established practices, our inversion network is implemented by a multi-layer MLP, which is a simple yet efficient design widely adopted in related works[1, 2]**.
> Existing literatures [1, 2] have demonstrated that such inverted pseudo-words effectively encapsulate both holistic visual content and intricate details, including implicit semantic attributes.
> More importantly, we fully agree that it's necessary to further strengthen its explicit semantic modeling for representing complex identities. Accordingly, our future improvements will include:
>
> - **1)** We plan to incorporate a lightweight attribute constraint module that aligns MLP outputs with a predefined semantic dictionary (covering "texture", "material," etc.) via contrastive loss;
>
> - **2)** We plan to add a cross-attention layer between structural features and attribute embeddings to enhance the inversion network’s ability to map fine-grained visual cues to corresponding textual semantics.
>
> These improvements will address the semantic deficiency concern while maintaining the model’s efficiency. We will elaborate on these in the revision.
>
> **References**:
>
> [1] "An image is worth one word: Personalizing text-to-image generation using textual inversion."  In International Conference on Learning Representations (ICLR), 2022.
>
> [2] "A pedestrian is worth one prompt: Towards language guidance person re-identification." Proceedings of the IEEE/CVF Conference on Computer Vision and Pattern Recognition. 2024.
>
> ---
>
> **Q1:  Modality prompt variants.**
>
> We sincerely apologize for the confusion regarding the design of modality prompt.
> In fact, the L-dMMC module does not use fixed modality prompts but instead introduces **learnable modality prompts** tailored to each modality.
> These prompts encode modality-specific information for missing modalities and, combined with the inverted identity pseudo-word, construct personalized linguistic description for each unavailable modality.
> They are dynamically updated through memory-guided optimization, enabling adaptive adjustments to diverse missing scenarios. To validate the effectiveness of our prompt design, we conducted ablation experiments on modality prompt variants, as shown Table below.
>
> |Variants|Mean mAP|Mean R-1|
> |:------------|:-----:|:-----:|
> | T-A (*w/o* Modality Prompt) |     53.9 |   54.8   |
> | T-B (Fixed Modality Prompt) |     54.0 |  55.0   |
> | T-C (Learnable Modality Prompt) | 54.6  | 55.7|
>
> - **T-A** ('_An image of a [Inverted Pseudo-word] person._') omits modality prompts entirely, relying only on identity pseudo-words;
>
> - **T-B** ('_An image of a [Inverted Pseudo-word] person in visible/near infrared/thermal infrared spectrum._') uses fixed descriptive phrases (e.g., "in visible spectrum") tied to modalities;
>
> - **T-C** ('_An image of a [Inverted Pseudo-word] person, who shows the [Learnable Modality Prompt] attributes._') employs learnable prompts that adaptively capture modality-specific semantics.
>
> The performance of T-A shows modality prompts are critical for cross-modal alignment. The weak improvement by-B confirms modality-specific guidance helps, but rigid templates lack contextual awareness. T-C outperforms T-A and T-B, confirming learnable prompts’ superiority.
>
> **The rationale behind learnable modality prompts is threefold**: 1) They dynamically adapt to pre-trained VLMs’ semantic space, aligning better with VLM’s inherent knowledge than rigid fixed phrases.
> 2) They adaptively encode modality-specific nuances (e.g., RGB’s color cues, thermal’s temperature features) that fixed prompts fail to capture, enhancing compensation precision for missing modalities.
> 3) Combined with inverted identity pseudo-words, they achieve dual personalization (modality + identity), strengthening fine-grained identity discrimination critical for ReID.
>
> We will elaborate on these details and add more analysis in the revision.
>
> ---
>
> Sincere thanks for your valuable insights, which have been instrumental in enhancing our work. We hope our replies fully address the raised issues, and we’d be honored by your backing of our submission. Appreciate your time and expertise.

---

> ### Comment · Area_Chair_jsjU · 2025-08-05
>
> Reviewer zX11, please engage in the discussion period. I understand that the review period started over a weekend, but we only have a few days remaining in the (now slightly extended) discussion period. The author's have provided a thoughtful response to your review and you are obligated to respond to it. You should share with the authors if they addressed questions or concerns you had, and seek clarification about any questions or concerns that remain. Please post your response as soon as you can so that there is time for the authors to follow up and discussion to progress as needed.

---

> > ### Comment · Reviewer_zX11 · 2025-08-06
> >
> > Thanks to the authors for the response. Some of my questions have been resolved. Therefore, I will maintain my original score.

---

> > > ### Author Response · Authors · 2025-08-06
> > >
> > > We are pleased that our rebuttal has addressed your concerns.
> > >
> > > Once again, we would like to express our sincere appreciation for your support of our work.

---

### Official Review · Reviewer_pLpE · 2025-07-02

**Clarity:** 3
**Significance:** 2
**Originality:** 3
**Rating:** 4
**Confidence:** 4

**Summary:**

This paper addresses the challenge that existing multi-modality ReID models degrade significantly when handling modality-incomplete inputs, proposing a flexible framework called Miss-ReID—the first to support both modality-missing training and inference. Experiments validate its superiority over state-of-the-art methods in various modality-missing scenarios, advancing multi-modality ReID toward real-world applications.

**Questions:**

1.	The novelty of the proposed method is not well-explained. It doesn't mean that this paper lacks novelty. I hope the authors can explain more details about the motivation of model design.

2.	How much is the performance difference when different hyper-parameters are used in the overall objective function?

3.	There are many sota methods (such as [1], [2], and [3]) in the field of person ReID. Why aren’t they adopted for comparison?

4.    This paper is well presented, but it still lacks significant contributions. If the authors can well solved my questions, I will raise my rating to more positive one.


[1] Yu, Chenyang, et al. "TF-CLIP: Learning text-free CLIP for video-based person re-identification." Proceedings of the AAAI conference on artificial intelligence. Vol. 38. No. 7. 2024
[2] Han, Qianru, et al. "CLIP-SCGI: Synthesized Caption-Guided Inversion for Person Re-Identification." arXiv preprint arXiv:2410.09382 (2024).
[3] Li, Siyuan, Li Sun, and Qingli Li. "Clip-reid: exploiting vision-language model for image re-identification without concrete text labels." Proceedings of the AAAI conference on artificial intelligence. Vol. 37. No. 1. 2023.

**Ethical Concerns:**

["Major Concern: Improper research involving human subjects"]

**Final Justification:**

The response has solved my concetns. I keep my rating.

**Limitations:**

The adopted datasets are previously proposed. They may not be suitable for the proposed miss-modality setting.

More sota methods should be cited for comparison though they are not proposed for solving the miss-modality problem.

**Quality:**

2

**Strengths And Weaknesses:**

S1: The writing of this manuscript is easy to understand and follow. Figure 2 explain the pipeline of the proposed method well. All implementation details are shown in this paper.

S2: The proposed setting in this paper is interesting, and it is meaningful for practical applications. The experimental results can well validate the effectiveness of the proposed method for person ReID.

S3: Through experimental results, the design of several modules (such as M-bHIPR) is helpful for this task. However, the reasons for designing these modules are not well presented in this paper.


W1: The visual encoder is initialized by the pre-trained model, such as CLIP, which may be unfair for compared methods. In addition, the authors do not claim the adopted model type of visual encoder in CLIP in the main text.

W2：The reserved features of previous training steps may influence the final performance. They may harm the performance of the model rather than being helpful for prediction.

W3: Too many loss items are introduced, resulting that it is hard to balance them in the overall objective function.

W4: It will be better if a comprehensive dataset can be introduced for the proposed miss-modality setting. This paper only utilizes the off-the-shelf datasets for experiments, which cannot well validate the effectiveness of the proposed method

---

> ### Author Rebuttal · Authors · 2025-07-29
>
> We deeply appreciate your constructive and insightful comments. Your support means a great deal to us, and thank you for recognizing our work！Below is our point-by-point response to all Weaknesses (W), Questions (Q), and Limitations (L).
>
> ---
> **W1: The details of pre-trained model.**
>
> - _**Fairness of using pre-trained VLM**_: CLIP has been widely adopted as a pre-trained backbone in various multi-modality tasks, including multi-modality ReID, due to its strong cross-modal alignment capability. Notably, one listed state-of-the-art method IDEA [25'CVPR], also utilizes CLIP as its pre-trained model. Despite sharing the same pre-trained foundation, our Miss-ReID still achieves significantly superior performance over IDEA in multiple modality-missing scenarios, e.g., `+3.7%` mAP when RGB is missing (NT) and `+4.8%` mAP when Thermal is missing (RN). These results clearly demonstrate that the advantages of Miss-ReID stem from our framework design (text-to-vision compensation mechanism) rather than the pre-trained model itself.
>
> ||Backbone|NT|RT|RN|T|N|R|Mean mAP|
> |:------------:|:-----:|:-----:|:-----:|:-----:|:-----:|:-----:|:-----:|:-----:|
> | IDEA|CLIP(ViT-B/16)|62.9 |71.5|58.4|43.3|27.1|39.9|50.5|
> | Miss-ReID|CLIP(ViT-B/16)|66.6|72.4|63.2|47.2|34.5|43.9|54.6|
>
> - _**Visual encoder type**_:  Regarding the visual encoder in CLIP, we use the widely adopted `ViT-B/16`, which is a standard choice in many related works and ensures consistency with common experimental setups in the field. We apologize for not explicitly stating this in the manuscript and will add this detail in the revised version to enhance clarity. To ensure reproducibility, we will release our code with well-trained models.
>
> ---
> **W2: The impact of reserved features in previous training steps.**
>
> We thank you for this valuable concern.
> The reserved features in our memory banks serve as heterogeneous prototypes for each identity, preserving multi-modality characteristics to guide pseudo-text generation when modalities are missing.​ To address potential performance interference from stale features, we employ two solutions as follows.
>
> - _**Dynamic update mechanism**_: Our memory banks first adopt a sliding average update mechanism.
> This dynamic update filters outdated prototypes by iteratively integrating new feature information, ensuring reliable and representative semantics are retained.​
>
> - _**Training phase separation**_: Notably, our model starts learning compensatory textual features under memory supervision only after 20 epochs.
> This delayed supervision allows our model to first capture stable visual representations, ensuring the memory-guided learning of textual features is both effective and robust.
>
> ---
> **W3/Q2: Hyper-parameter analysis ($\lambda_1$ and $\lambda_2$ in overall objective function Eq. 11).**
>
> We appreciate the your concern about loss balancing.
> We have conducted hyper-parameter analysis on the key loss weights in Eq. 11.
> Table below shows the performance differences under varying $\lambda_1$ (controlling triplet loss) and $\lambda_2$ (controlling memory-based contrastive constraint).
>
> | ($\lambda_1$, $\lambda_2$) | (0.1, 0.1) |  (0.1, 0.25) | (0.25, 0.5) | (0.5, 0.25) |
> |:-----:|:-----:|:-----:|:-----:|:-----:|
> | Mean mAP|  54.6 |  53.1 | 51.4 | 51.7
> | Mean R-1  |  55.7 |  53.8 | 52.1 | 52.4
>
> The results indicate that excessive weighting of triplet loss ($\lambda_1$) and memory contrast  ($\lambda_2$) impairs discriminative power, requiring careful calibration to avoid over-constraining the model.
> Our model with ($\lambda_1$=0.1, $\lambda_2$=0.1)  strikes this balance and yields optimal results.
> We will add these analyses in the revision to validate balance.
>
> ---
> **W4/L1: Customized modality-incomplete dataset.**
>
> We sincerely appreciate your constructive suggestion regarding dataset suitability.
>
> To simulate real-world modality-missing scenarios, we randomly mask images in different modalities at specific missing rates (0%-50%) on existing datasets (RGBNT201 and RGBNT100), following some practical causes.
> For example, RGB images are masked to mimic low-light/nighttime conditions where visual details are lost; thermal images are masked to simulate sensor failures in harsh environments.
> This approach ensures the simulated settings closely align with actual application bottlenecks, enabling valid evaluation of our method's robustness.
>
> - _**Future work**_: While current benchmarks validate core mechanisms, we fully agree that developing a dedicated dataset for modality-incomplete scenarios will strengthen real-world relevance. As you suggested, we plan to develop a comprehensive dataset tailored for the modality-missing setting.
> Our planned benchmark will cover more modality types (e.g., event/LiDAR data, sketches), establish the unified evaluation criteria, and include diverse real-world missing patterns: **1) Environmental-caused** (e.g., fog blocking RGB, rain interfering with depth sensors); **2) Hardware-caused** (e.g., temporary failure of thermal cameras); **3) Scene-caused** (e.g., partial occlusion hiding specific modalities).
>
> We believe it is our responsibility to contribute to this field, as it will facilitate more rigorous model validation and accelerate the practical deployment of multi-modality ReID systems.
>
> ---
> **Q1: Detailed motivations behind model design.**
>
> We appreciate this valuable feedback and elaborate on the design motivations below to better highlight the novelty.
> The core novelty of Miss-ReID lies in improving model's discriminative power in multi-modality ReID when arbitrary modalities are missing. To this end, three modules (M-bHIPR, M-iOSM and L-dMMC) are collaboratively designed to compensate the information absence based on the following motivations.
>
> - _**Memory-based Heterogeneous Identity Prototype Representation (M-bHIPR)**_:
> To provide supervision for aligning compensated textual features in missing modalities, we construct modality-specific memory banks that store heterogeneous prototypes for each identity, ensuring rich multi-modality cues for later compensation.
>
> - _**Modality-invariant Object Structure Modeling (M-iOSM)**_: The structural patterns encode identity-specific information that is preserved regardless of the sensory modality. Therefore, we extract modality-invariant structures via structure-aware queries, capture stable identity traits (e.g., body shape) that generalize across missing modalities.
>
> - _**Language-driven Missing Modality Completion (L-dMMC)**_: By inverting structural features into identity pseudo-word embeddings that encode identity semantics and incorporating learnable modality prompts, we generate synthetic textual descriptions. Leveraging VLMs' inherent vision-text alignment, these descriptions serve as reliable substitutes for missing visual cues, optimized via memory-based contrastive constraints to ensure consistency with visual features.
>
> We will expand these motivations in the revision for clarity.
>
> ---
> **Q3/L2: Comparisons with more sota methods.**
>
> We sincerely apologize for the incomplete comparisons.
>
> In the manuscript, we have compared our method with several state-of-the-art open-source models (e.g., DeMo [25'AAAI], IDEA [25'CVPR]) and demonstrated that our Miss-ReID achieves superior performance in diverse modality-missing scenarios.
>
> While acknowledging the excellent performances of TF-CLIP [1], CLIP-SCGI [2], and Clip-reid [3], it is important to note that they primarily focus on **single-modality ReID tasks** (video-based for [1], image-based for [2, 3]).
> In contrast, our work specifically targets multi-modality ReID with modality missing, which requires compensating and integrating complementary information across diverse modalities.
> *These single-modality methods lack the capability to handle heterogeneous multi-modality characteristics and thus cannot effectively cope with modality-missing scenarios.* A direct comparison would be unfair as the task settings and core challenges differ fundamentally.
>
> Nevertheless, we recognize the value of broader benchmarking and will incorporate additional SOTA methods (including the mentioned works) in the revised manuscript to strengthen the empirical evaluation.
>
> ---
> **Q4: Our contributions.**
>
> We appreciate your valuable feedback, and our distinct contributions can be further highlighted as follows:
>
> - _**Conceptual novelty**_: Miss-ReID is **the first framework** to enable modality-missing training and inference for multi-modality ReID. Unlike most existing methods that assume complete modalities during both stages, our approach natively supports arbitrary modality-missing inputs whether during training (e.g., partial modality data for certain identities) or inference (e.g., sensor failures in deployment). This flexibility preserves multi-modality representation power while adapting to incomplete data streams, **a key step toward practical multi-modality ReID systems**.
>
> - _**Methodological innovation**_: we pioneer using Vision-Language Models (VLMs) for dynamic textual compensation in modality-missing ReID. Instead of relying solely on image or visual feature compensation, we leverage VLMs’ inherent vision-text alignment to distill structural semantics from available modalities into semantic-aligned textual embeddings.
> This novel **"Text-to-Visual Compensation"** mechanism, combined with our memory banks, structure-aware query interactions, and modality-adaptive prompts, enables robust retrieval even when critical visual cues are absent.
> **It highlights VLMs as an underexplored tool for handling incomplete multi-modality data in ReID**.
>
> We will strengthen these points in the revision to better highlight our contributions.
>
> ---
> Thank you sincerely for your insights, which have significantly helped refine our work.
> We hope our responses address all concerns adequately, and we would be grateful for your consideration in supporting our submission. Grateful for your time and expertise.

---

> > ### Comment · Reviewer_pLpE · 2025-08-05
> > **Thanks for your response**
> >
> > Thanks for your response. I have read all your response. The supplemented results have solved my concerns. They should be added in the final revision. I will keep my positive rating.

---

> > > ### Author Response · Authors · 2025-08-05
> > >
> > > Again, we’re truly grateful for your support of our work and the positive rating!
> > >
> > > We will promptly incorporate the supplemented results into the final revision to ensure its completeness and rigor.

---

> ### Comment · Area_Chair_jsjU · 2025-08-05
>
> Reviewer pLpE, please engage in the discussion period. I understand that the review period started over a weekend, but we only have a few days remaining in the (now slightly extended) discussion period. The author's have provided a thoughtful response to your review and you are obligated to respond to it. You should share with the authors if they addressed questions or concerns you had, and seek clarification about any questions or concerns that remain. Please post your response as soon as you can so that there is time for the authors to follow up and discussion to progress as needed.

---

### Official Review · Reviewer_z2ZA · 2025-07-03

**Clarity:** 2
**Significance:** 3
**Originality:** 2
**Rating:** 4
**Confidence:** 4

**Summary:**

This paper presents a flexible framework tailored for a more realistic multi-modality retrieval scenario, dubbed Miss-ReID. Miss-ReID aims at compensating for missing visual cues via vision-text knowledge transfer driven by Vision-Language foundation Models (VLMs), effectively mitigating performance degradation.

**Questions:**

1. Both [18] and [32] address modality-missing scenarios during training/inference. Concretely clarify the fundamental distinctions between Miss-ReID and these works, and supplement comparative experiments.
2. The M-iOSM module assumes cross-modal invariance of object structures, yet significant contour discrepancies may occur under low-light conditions (e.g., RGB vs. TIR). How to guarantee truly modality-invariant structural features? Could this introduce noise?
3. Current experiments only simulate random modality missing. How would Miss-ReID maintain robustness when encountering inter-modal dependent missing (e.g., NIR invariably missing when RGB is unavailable)?
4. As shown in Table 2, Miss-ReID’s mAP under full modalities (RNT) (76.9%) lags behind DeMo (79.0%) and IDEA (80.2%). Does the textual compensation module introduce interference to visual features when modalities are complete?  How to optimize this?
5. L-dMMC increases FLOPs by 23% (Table 1). Are there plans to develop a lightweight variant(e.g., via prompt distillation) for edge devices? What is the actual inference speed?

**Ethical Concerns:**

["NO or VERY MINOR ethics concerns only"]

**Final Justification:**

The author's response has addressed my questions. I am willing to raise my score.

**Limitations:**

yes

**Quality:**

2

**Strengths And Weaknesses:**

Strengths
1.  Miss-ReID handles multi-modality ReID under more general modality-missing scenarios encountered during both training and inference. Miss-ReID allows the arbitrary modality-missing inputs, while preserving the multi-modality representation capacity.
2. Bolstered by the inherent vision-text reasoning capabilities of Vision-Language foundation Models (VLMs), Miss-ReID dynamically compensates for missing visual cues through semantic-aligned
textual embeddings, and our intriguing findings highlight the potential of developing VLMs within the realm of multimodal ReID, encountering incomplete data streams.
3. Comprehensive experiments underscore the model’s efficacy and superiority over state-of-the-art methods in various modality-missing retrieval scenarios
Weakness:
1. The paper claims to be the "first work to handle modality-missing scenarios in both training and inference" (intro). However, prior works like Lee et al. (2023) [18]  and Ke et al. (2025) [32] have addressed missing modalities in both phases—e.g., via learnable modality-missing-aware prompts or LMM-based completion. The distinction between Miss-ReID and these methods requires clearer articulation to justify its "first work" claim.
2. The structure-aware querying mechanism lacks critical implementation specifics, including undefined initialization/optimization objectives for learnable vectors Q, no explicit loss (e.g., cross-modal alignment) ensuring structural consistency across modalities (e.g., RGB-TIR feature alignment), and absent quantitative validation of said consistency.
3. The L-dMMC module increases FLOPs by 27% (34.3G → 43.6G, Table 1) but omits: Inference latency metrics (critical for real-world deployment, e.g., surveillance systems)
4. Textual compensation efficacy lacks fine-grained validation in terms of VLM backbone comparisons (e.g., CLIP vs. BLIP-2) or template designs. At the same time, the contribution of pseudo-word generation is only qualitatively demonstrated (Fig. 6), lacking quantitative ablation studies, such as performance drops without the inversion network.

---

> ### Author Rebuttal · Authors · 2025-07-29
>
> We wish to convey our sincere thanks for your valuable comments. Your support means a great deal to us, and we would be truly honored if you would act as an advocate for our work.
> Below is our comprehensive, point-by-point response to all the identified Weaknesses (W) and Questions (Q).
>
> ---
>
> **W4/Q1: About the description of ''the first work''.**
>
> We sincerely apologize for the insufficient clarity in our original description.
> Strictly speaking, our "first work" claim specifically refers to the **multi-modality ReID task**, where no prior work has addressed modality missing in both training and inference.
> We sincerely acknowledge that [18] and [32] have explored modality-missing scenarios across training and inference, but their focus differs fundamentally from ours.
>
> Specifically, [18] and [32] primarily address tasks like multi-modal classification or image/text generation, where the core challenge lies in **label prediction or content synthesis**. In contrast, our work targets multi-modality ReID, as a retrieval task requiring **fine-grained identity discrimination**, which demands capturing subtle inter-identity differences rather than coarse-grained classification/generation. We will add these distinctions into the Related Work in the revision.
>
> Given the task and technical disparities, we are actively adapting [18] and [32] to the multi-modality ReID setting (e.g., modifying their frameworks for identity retrieval) and will include comparative experiments in the revision for comprehensive benchmarking.
>
> ---
>
> **W5/Q2:  Some issues about structure-aware querying mechanism in M-iOSM module.**
>
> We sincerely apologize for unclear descriptions about structure-aware querying mechanism, and our detailed responses are as follows.
>
>   _**a. Optimization objective for learnable vectors Q**_: Learnable vectors Q are optimized via two main types of constraints: 1) **Direct** identity loss and triplet loss ensure structural features are identity-discriminative. 2) **Indirect** text-image contrastive loss from L-dMMC ensures that the structural features, when transformed into pseudo - words through the inversion network, can effectively encapsulate the visual content. The combination of these two types of constraints guarantees that the structural features **not only carry unique identity information** but also **maintain a strong connection with the corresponding textual descriptions**, facilitating cross-modal understanding.
>
>   _**b. Structural consistency constraints**_: Due to imaging technologies or conditions, a small number of multi-modality images may have subtle structural differences or modality-specific noise (e.g., TIR thermal blur). **Forcing the structural features to be strictly consistent may instead lead to the model's overfitting on noisy. To avoid this problem, we eschew rigid alignment by averaging structural features across different modalities to form the final structural representation.**
> This "consensus-based" approach mitigates outlier impacts and leverages cross-modality complementarity, maintaining the stability and reliability of the overall structural feature representation.
>
>   _**c. Reliability of structure features obtained under low-light conditions**_: As mentioned above, our module mitigates the impact of low-quality modalities by averaging the structural features from different modalities. However, further improvements are still needed. As future work, the improvements will be as follows:
>
>  - We plan to employ some pre-processing techniques specifically designed for low-light images. These can boost visual and semantic features while preserving details in low-light images, improving input quality for M-iOSM to enhance structural feature reliability.
>
> - We plan to integrate attention mechanisms into M-iOSM, such as multi-scale attention. This will help the model focus on reliable structural cues, reduce noise/inconsistency impacts (especially in low light), and adaptively weight feature parts to prioritize identity-discriminative regions.
>
> We will elaborate on these details and add more analysis in the revision.
>
> ---
>
> **W6/Q5: Model complexity and inference speed.**
>
> We appreciate your valuable concerns about computational complexity and inference speed, which are critical for real-world deployment. As suggested, we’ve added detailed comparisons of model complexity and inference speed between Miss-ReID and SOTA methods, along with in-depth analyses.
> _However, due to the 10,000-character limit, the detailed response to this issue can be found in Reviewer uuXE (Q1)._
> **We sincerely apologize and appreciate your understanding.**
>
> ---
>
> **W7:  Some supplementary ablation studies.**
>
>   _**a. CLIP vs. BLIP-2**_:
> We have evaluated various VLMs as backbones (**with only text encoders frozen during training**), and the results are presented in Table below.
> |Backbone| Image Encoder| Text Encoder| Params (Trainable)| Params (Total)| Mean mAP|Mean R-1|
> |:------------:|:-----:|:-----:|:-----:|:-----:|:-----:|:-----:|
> | CLIP    |  ViT-B/16  | Transformer| 89.6M   |   153.0 M   |  54.6  | 55.7|
> | BLIP-2 |  ViT-L/14  | OPT-2.7B    | 310.6M |      3.1B     |  55.4  | 56.4 |
>
> The table shows BLIP-2, with 3.4× more trainable params and 20× total params than CLIP, yields marginal gains (mAP +0.8%, R-1 +0.7%), indicating stronger VLMs don’t guarantee proportional improvements.
> This may due to that BLIP-2’s 3.1B params need larger data to realize its potential, and BLIP-2 excels at generative tasks, while CLIP’s contrastive pretraining better fits cross-modality retrieval, offering efficient features for ReID.
>
>
> _**b. Template design**_:
> To validate the effectiveness of the template design, we have conducted ablation experiments on template variants, as shown in Table below.
>
> |Variants|Mean mAP|Mean R-1|
> |:------------|:-----:|:-----:|
> | T-A (*w/o* Modality Prompt) |     53.9 |   54.8   |
> | T-B (Fixed Modality Prompt) |     54.0 |  55.0   |
> | T-C (Learnable Modality Prompt) | 54.6  | 55.7|
>
> - T-A ('_An image of a [Inverted Pseudo-word] person._') omits modality prompts entirely, relying only on identity pseudo-words;
>
> - T-B ('_An image of a [Inverted Pseudo-word] person in visible/near infrared/thermal infrared spectrum._') uses fixed descriptive phrases (e.g., "in visible spectrum") tied to modalities;
>
> - T-C ('_An image of a [Inverted Pseudo-word] person, who shows the [Learnable Modality Prompt] attributes._') employs learnable prompts that adaptively capture modality-specific semantics.
>
> The performance of T-A shows modality prompts are critical for cross-modal alignment. The weak improvement by-B confirms modality-specific guidance helps, but rigid templates lack contextual awareness. T-C outperforms T-A and T-B, confirming learnable prompts’ superiority. This result indicates the modality-specific learnable prompts dynamically adapt to pre-trained VLMs’ semantic space, aligning better with VLM’s inherent knowledge than rigid fixed phrases.
>
> _**c. Contribution of Inversion Network**_:
> We apologize for the lack of quantitative analysis on the inversion network. We have conducted ablation experiments comparing performance with and without the inversion network, as reported in Table below.
>
> |Variants|Mean mAP|Mean R-1|
> |:------------|:-----:|:-----:|
> | *w/o* Inversion Network  | 53.7  | 54.8|
> | w/      Inversion Network  | 54.6  | 55.7|
>
> The result shows that removing the inversion network leads to certain performance drop, underscoring its necessity in **bridging visual-textual gaps**. It can be seen that, by converting structural visual cues into textual embedding, it enables more precise cross-modality alignment, enhancing the L-dMMC module’s ability to generate identity-specific compensation semantics.
>
> We will supplement these ablation studies with more analysis in the revision.
>
> ---
>
> **Q3: The robustness of model when encountering inter-modal dependent missing.**
>
> We appreciate this insightful question. While current experiments primarily simulate random modality missing, they do cover diverse missing combinations including simultaneous RGB/NIR failure at night.
> Our Miss-ReID enhances discriminative ability by generating textual features for missing NIR and RGB, to some extent.
> To better address inter-modality dependent missing, future work will create real modality-incomplete datasets (extensively covering such correlated patterns as suggested) and design adaptive fusion modules to explicitly model cross-modality failure correlations, ensuring more reliable feature compensation.
>
> ---
>
> **Q4:  Does the textual compensation module introduce interference to visual features when modalities are complete?**
>
> We sincerely apologize for our unclear description about Table 2. We clarify that Miss-ReID's textual compensation module (L-dMMC) selectively activates only for missing modalities during inference. The 76.9 mAP in Table 2 refers to **visual-only performance** under full modalities (RNT). We also supplement comparative results with/without using textual features in the Table below.
>
> ||mAP (complete)|Mean mAP (missing)|
> |:-----:|:-----:|:-----:|
> | DeMo |  79.0    |  50.7 |
> | IDEA   |  80.2    |  50.5 |
> | Miss-ReID (*w/o* text) |  76.9    |  53.3 |
> | Miss-ReID (*w/* text)  |  77.7|  54.6|
>
> The results indicate that adding textual features improves mAP to 77.7%, confirming textual features do not interfere but enhance discriminability in modality-complete inference.
> Moreover, DeMo and IDEA excel in complete modalities thanks to their complex multi-modality interaction designs, **which heavily depend on on modality-complete inputs**, rendering them fragile when modalities go missing.
> We will add these details in the revision.
>
> ---
>
> We’re deeply grateful for your insightful feedback, which has significantly strengthened our work.
> We trust our responses have thoroughly addressed your concerns, and we’d be privileged to have your support for our submission. Thank you for your time and expertise.

---

> ### Comment · Area_Chair_jsjU · 2025-08-05
>
> Reviewer z2ZA, please engage in the discussion period. I understand that the review period started over a weekend, but we only have a few days remaining in the (now slightly extended) discussion period. The author's have provided a thoughtful response to your review and you are obligated to respond to it. You should share with the authors if they addressed questions or concerns you had, and seek clarification about any questions or concerns that remain. Please post your response as soon as you can so that there is time for the authors to follow up and discussion to progress as needed.

---

### Decision · Program_Chairs · 2025-09-17

**Decision:**

Accept (poster)

**Comment:**

The paper presents Miss-ReID, a framework for multi-modality object re-identification under missing modality conditions during both training and inference, using vision-language models to compensate for absent visual inputs.

Reviewers thought the paper addresses a realistic and underexplored problem setting, presents a technically coherent and modular architecture, and demonstrated strong empirical results. They also thought the paper was well written and had thorough ablation studies.Concerns raised in the initial reviews included lack of clarity in the novelty claim versus prior work, limited exploration of VLM variants, and architectural details around structure modeling and inversion mechanisms. However, most of these were effectively addressed in the rebuttal and discussion, and reviewers seemed satisfied.

Given the sound technical contributions, practical significance, and resolution of key concerns during the discussion, as well as the final ratings of accept and three borderline accepts, the AC recommends acceptance.